# Simulation-Based Inference for Adaptive Experiments

**Brian M Cho**
Cornell Tech
bmc233@cornell.edu

**Aurélien Bibaut**
Netflix
abibaut@netflix.com

**Nathan Kallus**
Netflix & Cornell Tech
kallus@cornell.edu

## Abstract

Multi-arm bandit experimental designs are increasingly being adopted over standard randomized trials due to their potential to improve outcomes for study participants, enable faster identification of the best-performing options, and/or enhance the precision of estimating key parameters. Current approaches for inference after adaptive sampling either rely on asymptotic normality under restricted experiment designs or underpowered martingale concentration inequalities that lead to weak power in practice. To bypass these limitations, we propose a simulation-based approach for conducting hypothesis tests and constructing confidence intervals for arm specific means and their differences. Our simulation-based approach uses positively biased nuisances to generate additional trajectories of the experiment, which we call *simulation with optimism*. Using these simulations, we characterize the distribution potentially non-normal sample mean test statistic to conduct inference. We provide guarantees for (i) asymptotic type I error control, (ii) convergence of our confidence intervals, and (iii) asymptotic strong consistency of our estimator over a wide variety of common bandit designs. Our empirical results show that our approach achieves the desired coverage while reducing confidence interval widths by up to 50%, with drastic improvements for arms not targeted by the design.

## 1 Introduction

In recent years, adaptive experimental designs have gained increasing popularity over the classic randomized controlled trial. Bandit algorithms, where treatment assignment probabilities are updated sequentially and simultaneously with data collection, are increasingly used for objectives such as welfare maximization during experimentation [1, 20], quickly identifying well-performing option(s) [8, 11], and maximizing power against particular hypotheses [16, 25, 37]. Compared to classic fixed designs, modern approaches offer the promise of flexible, efficient experimentation by leveraging information collected *during* the experiment.

However, once the experiment is over, researchers are still interested in using adaptively collected data to conduct inference on a variety of different quantities, including those not targeted by the design. For example, while an adaptive design may sample the best-performing option at a higher rate, experimenters may still be interested in conducting inference on all offered options in the experiment. Policymakers may be interested in whether the best-performing option outperforms other alternatives with a certain level of statistical significance to assess the credibility of their conclusions. Such goals necessitate after-study inference. However, while adaptive experiments offer numerous benefits, they pose challenges for after-study inference [2, 22]. Unlike classic randomized controlled trials, standard hypothesis tests and confidence intervals are not guaranteed to provide their nominal error control

(e.g., contain the true target of inference with a pre-specified desired error probability) due to the dependence across observations induced by adaptive sampling

Existing works addressing this issue primarily rely on two distinct approaches. Some works [4, 13] propose reweighing approaches in order to enforce asymptotic normality and use Wald-style confidence intervals. These approaches, however, restrict the class of experiment designs. Sampling schemes must have nonzero probability of selecting an option at each timestep of the trial, which excludes popular designs such as explore-then commit (ETC) and upper-confidence-bound style (UCB) algorithms. As an alternative to asymptotic normality, anytime valid inference approaches [5, 7, 8, 24, 32, 33] ensure correct error control across all adaptive designs. These approaches, however, are often underpowered in many use cases due to protecting for all possible sampling schemes, rather than the exact design used for data collection.

**Contributions.** We provide a novel asymptotic inference approach for adaptive experimental designs. Our approach relies on a simulation procedure that adds positive bias towards estimated nuisances, which we call *simulation with optimism*. Our procedure conducts hypothesis tests for values of arm means and their differences using these simulated distributions, providing natural confidence intervals and point estimates. We prove that our approach maintains desired type I error control over a wide class of commonly used designs, *including designs which violate the conditional positivity assumption* necessary for asymptotic approaches. Crucially, we show the benefits of our approach on both synthetic data and real-world data collected adaptively from the Amazon MTurk platform. Across experiments, our method demonstrate improvements in interval widths and estimation error *by up to 50%*, with dramatic improvements for parameters not specifically targeted by the design.

**Outline.** In the remainder of this section, we provide a brief overview of inference approaches for data collected under adaptive designs. In Section 2, we introduce the notation and set-up for our problem. In Section 3, we introduce our algorithm. We provide intuition for our algorithm using a simple ETC example, and show our approach protects type I error under multiple designs commonly used in practice. In Section 4, we test our approach on both synthetic setups and real-world data collected from a political science survey experiment, showing that our approach achieves tighter confidence intervals while preserving Type I error control.

## 1.1 Related Works

**Reweighing for Asymptotic Normality.** Many existing works aim to provide valid inference and hypotheses tests by reweighing existing estimators [6] for a fixed sampling scheme and stopping time. These reweighing schemes [4, 13, 35] aims to stabilize variances and recover the conditions of a martingale central limit (MCLT) theorem [14], enabling standard Wald-style confidence intervals based on asymptotic normality. Popular approaches [13] propose weighing schemes that heuristically aim to reduce variance, under variance convergence conditions sufficient for ensuring asymptotic normality. These approaches rely on strict rates of exploration for asymptotic normality to hold. Specifically, at each timestep, the design must have nonzero probability of selecting a given arm, conditional on the observed history. This is violated even in the simplest of adaptive sampling schemes, such as ETC and UCB. To provide inferential guarantees without such restrictions, other works have proposed inferential tools that satisfy *anytime validity*, which provides valid inference across all potential experiment designs.

**Anytime Valid Inference.** Anytime valid inference [15, 24] sidesteps asymptotic normality altogether in order to provide inferential guarantees under any experiment design. Nonasymptotic anytime valid inference rely on martingale concentration inequalities [30] to obtain their guarantees. However, these approaches require known bounds on the moment generating function of the underlying distribution [15]. To improve power and sidestep these limitations, other works have provided anytime valid approaches with *asymptotic guarantees* [5, 32], which rely on invariance principles [21] to obtain their approximate guarantees. While these approaches provide better power empirically than exact approaches, asymptotic anytime valid approaches are often still conservative due to protecting for *all* potential designs after a burn-in period, rather than the experiment design used in the study.

**Existing Simulation-Based Approaches.** Instead of enforcing asymptotic normality or providing anytime valid guarantees, our approach leverages a generative simulation procedure to approximate the distribution of our test statistic. Due to each observation depending on the entire history, bootstrap and block bootstrap approaches [10, 28] are not directly applicable to adaptively collected data.

Previous works in this setting focus specifically on tractable experiment designs, such as play-the-winner [34] sampling algorithm, in order to obtain analytical limiting distributions for inference. Other works [26] leverage a bootstrap procedure using an estimate of the data-generating distribution in the case of Bernoulli data. Our approach most closely aligns with the latter approach. However, our method can (i) handle more than two arms, (ii) accommodates nonparametric arm distributions, and (iii) applies to a wide class of adaptive experiment designs commonly used in practice.

## 2 Notation and Set-up

Throughout this work, we denote random vectors and random variables as $\mathbf{X}$ and $X$ in uppercase. We denote realizations of random vectors and random variables as $\mathbf{x}$ and $x$ in lowercase. We use the set $[N]$ to denote the set of integers $\{1, ..., N\}$, and use the subscripts $X_t$ as the time index for $t \in \mathbb{N}$. We use $\theta^*, \eta^*$ to denote the true underlying value of unknown parameters $\theta, \eta$ in the experiment.

We aim to analyze data sequences generated by interactions between an environment and an experimenter in the multi-armed bandit setting. We assume there exists $K$ treatments (or arms), each with a distribution $P_a$ corresponding to each arm $a \in [K] = \{1, ..., K\}$. Over successive rounds $t \in \mathbb{N}$, an action $A_t$ from an action set $[K]$ is selected, then generates an outcome $X_t \in \mathbb{R}$ according to a distribution $P_{A_t}$. The action $A_t$ is selected based on a known policy $\pi_t : H_{t-1} \to \Delta^K$, where $H_{t-1} = (A_i, X_i)_{i=1}^{t-1}$ denotes all observations up to time $t-1$ and $\Delta^K$ denotes the probability simplex over $[K]$. The experiment terminates at time $T$, resulting in an observed sequence $H_T = (A_i, X_i)_{i=1}^{T}$.

We denote the number of pulls and the sample mean of arm $a$ up to time $t \in [T]$ as $N_t(a) = \sum_{i=1}^{t} \mathbf{1}[A_t = a]$ and $\hat{\mu}_t(a) = \frac{\sum_{i=1}^{t} \mathbf{1}[A_t = a] X_i}{N_t(a)}$ respectively. We assume that the sampling scheme $\pi = \{\pi_t\}_{t \in T}$ is known, as is common in practice for adaptive experiments. Furthermore, we put the following assumptions on our adaptive design $\{\pi_t\}_{t \in [T]}$ and arm distributions $\{P_a\}_{a \in [K]}$.

**Assumption 1** (Infinite Sampling). *For each $a \in [K]$, $\lim_{T \to \infty} N_T(a)$ diverges to infinity almost surely for any set of arm distributions $\{P_a\}_{a \in [K]}$.*

This assumption is generally satisfied in practice, even in designs that aggressively aim to maximize the average outcome. Regret-optimal sampling schemes (i.e., sampling schemes that achieve the largest possible expected value within the trial duration) satisfy Assumption 1 by sampling all arms at least on the order of $\log(T)$ [19]. Assumption 1 does not imply conditions such as conditional positivity, where the probability of selecting an arm $a \in [K]$ must remain above zero for all $t \in [T]$.

**Remark 1** (Assumption of Known Policy). *We note that knowledge of the experiment policy is a standard assumption for after-study inference in adaptive experiments. Even when using reweighting/martingale methods, knowledge of the policy (i.e. full knowledge of conditional propensity scores at each time t) is required to implement existing inference approaches. Zhang et al. [36] use a reweighted estimator based on the martingale central limit theorem (MCLT) that directly leverages the true propensity scores in each term of the summation, and propose a weighting scheme that also incorporates the true propensity scores. Bibaut et al. [3] use a different reweighting approach for a similar inference approach based on the MCLT and still require knowledge of the true propensity scores at each time to construct their confidence interval. Lastly, Hadad et al. [13] build upon unbiased scoring rules that leverage the propensity scores in the denominator (similar to the augmented inverse propensity weighted (AIPW) approach), and assume that the conditional propensities (and therefore the sampling scheme) are known.*

### 2.1 Problem Statement

In this work, we focus on conducting inference on arm-specific means and their pairwise differences. We denote $\mu_a = \mathbb{E}_{P_a}[X]$ as the mean of arm $a$ for each arm $a \in [K]$, and use $\tau_{a,a'} = \mu_a - \mu_{a'}$ to denote the difference in means between arms $a$ and $a'$. We use $\theta$ to denote our target parameter (i.e., an arm-specific mean or their pairwise difference). Our goal is to conduct pointwise hypothesis tests for values of $\theta$ and construct confidence intervals for $\theta^*$ that provide asymptotic type I error control at level $\alpha$. We formally define asymptotic error control for our hypothesis tests and confidence intervals, starting with hypothesis tests in Definition 1.

**Definition 1** (Type I Error Control). *Let $\xi : (\theta_0, \alpha, H_T) \to \{0, 1\}$ be a test for null hypothesis $\theta = \theta_0$ with nominal Type I error probability $\alpha$. Let $\xi(\theta_0, \alpha, H_T) = 1$ denote rejection of the null*

$\theta = \theta_0$. *We say that the test $\xi(\theta_0, \alpha, H_T)$ has Type I error control if the probability of rejecting $\theta_0$ when $\theta^* = \theta_0$ is at most $\alpha$ as $T \to \infty$, i.e.*

$$\limsup_{T \to \infty} \mathbb{P}_{\theta^* = \theta_0} \left( \xi(\theta_0, \alpha, H_T) = 1 \right) \le \alpha. \tag{1}$$

Similarly, we define error control for confidence intervals with respect to the probability that a constructed interval does not contain $\theta^*$, the ground truth value of our target parameter of interest $\theta$.

**Definition 2** (Coverage of Confidence Set). *Let $C(\alpha, H_T)$ be a mapping from a prespecified $\alpha$-level and the observed data $H_T$ to an interval in $\mathbb{R}$. We say that the set $C(\alpha, H_T)$ has asymptotic coverage $1 - \alpha$ if $\theta^*$ is contained in the (random) confidence interval with at least probability $1 - \alpha$, i.e.*

$$\limsup_{T \to \infty} \mathbb{P} \left( \theta^* \notin C(\alpha, H_T) \right) \le \alpha \tag{2}$$

A test with type I error control directly leads to a confidence interval for the true value of $\theta$. By including all values of $\theta$ not rejected by a test with type I error control, one obtains a confidence set with the desired coverage level. In the following section, we provide our simulation-based test that satisfies the asymptotic error control condition in Definition 1 for common adaptive designs. By inverting this test, we construct asymptotically valid confidence intervals that satisfy Definition 2 and heuristic point estimates for our parameter of interest.

**Remark 2** (Why asymptotic control?). *One may ask why we seek asymptotic control, rather than control for any sample size $T \in \mathbb{N}$. We note that this is in line with standard approaches for statistical inference. Empirically well-powered approaches for inference with adaptively collected data [3, 5, 13] only offer asymptotic guarantees as in Definitions 1 and 2. Even in standard settings where observations are distributed independently, the most common mode of inference is the Wald confidence interval, which offers only asymptotic guarantees for type I error and coverage.*

## 3 Simulation Based Inference

To simplify exposition, we focus on arm-specific means as our target parameter, using $\mu_1$, the mean of arm 1, as our running example. We provide equivalent results for the difference-in-means target parameter in Appendix B. We first provide the pseudocode our simulation-based inference approach. We then discuss nuisance estimation, and introduce the principle of *simulation with optimism*. We demonstrate the importance of this principle with case study using a simple ETC design, and provide theoretical guarantees regarding type I error for designs used commonly in practice. Finally, we provide our confidence interval and point estimate that leverage our testing procedure. We show minimal convergence guarantees for both approaches, and discuss their practical considerations.

### 3.1 Pseudocode for our Approach

Our approach for conducting tests and constructing confidence intervals is based on a generative procedure for producing simulated experiment trajectories under the null hypothesis $\theta = \theta_0$. While the null hypothesis specifies the value of the target parameter, we require arm distributions and means for all other arms in order to simulate trajectories, which we refer to as *nuisance* parameters. We provide a simulation procedure in Algorithm 1, which generates observations from each arm with Gaussian noise[1]. Under the Gaussian noise trajectory simulation, we set our nuisances $\boldsymbol{\eta}$ to be the means of all arms other than target arm 1, as well as the variances for all arms. Note that we do not assume that the true underlying arm distributions follow a normal distribution.

Given our trajectory simulation procedure, we provide our approach for point null hypothesis testing in Algorithm 2. Our procedure first computes the sample mean test statistic $\rho(H_T) = \hat{\mu}_T(1)$ on the observed data $H_T$. We then generate $B$ trajectories of our experiment using Algorithm 1 to approximate the distribution of the sample mean test statistic under the null $\theta = \theta_0$. We reject the point null $\theta = \theta_0$ if the observed sample mean $\rho(H_T)$ falls below (or above) the lower (or upper) $\alpha/2$ quantile of the simulated distribution, resembling a standard two-sided test. In Algorithm 2, we return our decision to reject/accept the null $\theta = \theta_0$ and its corresponding $p$-value.

---

[1]In the setting of parametric arms (e.g., Bernoulli arms), where arm means specify each arm's distribution, the nuisance parameters $\eta$ for target parameter $\theta = \mu_1$ are just all other arm means $\boldsymbol{\mu}_{-1} = [\mu_2, \ldots, \mu_K]$.

---

**Algorithm 1** Trajectory Simulation

---

1: **input**: observed data $H_T$, sampling scheme $\pi$, point null value $\theta_0$.
2: Estimate nuisance parameters $\hat{\boldsymbol{\eta}} = [\hat{\mu}_2, ..., \hat{\mu}_K, \hat{\sigma}_1^2, ..., \hat{\sigma}_2^2]$.
3: Set $\hat{H}_0 = \{\emptyset\}$, and set $t = 0$.
4: **for** $t$ in $[T]$ **do**
5:      Sample $A_t$ according to the policy $\pi_t : \hat{H}_{t-1} \to \Delta^{[K]}$.
6:      Generate $X_t$ according to the normal distribution $N(\mathbf{1}[A_t \neq 1]\hat{\mu}_{A_t} + \mathbf{1}[A_t = 1]\theta_0, \hat{\sigma}_{A_t}^2)$.
7:      Set $\hat{H}_t = \hat{H}_{t-1} \cup (A_t, X_t)$.
8: **Return** $\hat{H}_T$.

---

**Algorithm 2** Point Null Testing via Resimulation

---

1: **input**: observed data $H_T$, sampling scheme $\pi$, type I error $\alpha$, sim. number $B$, null value $\theta_0$.
2: Estimate nuisance parameters $\hat{\boldsymbol{\eta}}$, and estimate observed test statistic $\rho(H_T) = \frac{\sum_{t=1}^T \mathbf{1}[A_t=1]X_t}{\sum_{t=1}^T \mathbf{1}[A_t=1]}$.
3: **for** $i$ in $[1, B]$ **do**
4:      Generate trajectory $H_T^{(i)}$ by resimulating the experiment according to Algorithm 1.
5:      Calculate the test statistic from the generated trajectory $\hat{\rho}^{(i)} = \rho(H_T^{(i)})$.
6: Denote the CDF of the simulated distribution as $\hat{F}(x) = \frac{1}{B}\sum_{i=1}^B \mathbf{1}[\hat{\rho}^{(i)} \leq x]$. Calculate $\hat{F}(\rho(H_T))$, the CDF of simulated test statistic evaluated at the observed test statistic $\rho(H_T)$.
7: **Return** $\left(1 - \mathbf{1}\left[\alpha/2 \leq \hat{F}(\rho(H_T)) \leq 1 - \alpha/2\right], \hat{F}(\rho(H_T))\right)$.

---

The choice of the sample mean test statistic $\rho(H_T) = \hat{\mu}_T(1)$ is motivated by minimal power results for our test. Specifically, under Assumption 1, the sample mean test statistic guarantees that Algorithm 2 rejects the nulls $\theta_0 \neq \theta^*$ as the trial duration grow large. We formalize this in Lemma 1 below.

**Lemma 1** (Test of Power 1.)**.** *Let $\theta = \theta_0$ be the point null, and assume Assumption 1 holds. Furthermore, assume that for all $a \in [K]$, the variances of distribution $P_a$ is finite. Then, if $\theta_0 \neq \theta^*$, the probability that Algorithm 2 rejects $\theta_0$ converges to 1 almost surely as $T \to \infty$.*

Lemma 1 is a direct consequence of the strong law of large numbers for sample means, which holds under the conditions of Assumption 1 even under adaptive designs. While asymptotic power is guaranteed by choosing the sample mean as our test statistic, obtaining valid type I error guarantees is obtained by our method of estimating nuisances $\hat{\boldsymbol{\eta}}$. By estimating nuisances *optimistically*, we show that our simulation-based test provides valid error control.

### 3.2 Simulating with Optimism

The estimation of nuisances $\boldsymbol{\eta}$ plays a crucial role in controlling the type I error rate. Even with simple adaptive designs in the two-armed case, simple plug-in nuisances do not provide type I error guarantees even if nuisances are estimated at parametric (i.e. $\Theta(1/\sqrt{T})$) rates. As a leading example of this phenomenon, we present a simple ETC design with two arms in Example 1.

**Example 1** (Two-Armed ETC)**.** *Let $K = 2$, and arm outcome distributions $P_1$, $P_2$ be any two distributions with finite variance. For $t \leq T/2$, let $\pi_t$ map to the uniform distribution over $[K]$. For $t > T/2$, let $\mathbb{P}_{\pi_t}(A_t = a) = 1$ for $a = \text{argmax}_a \hat{\mu}_{T/2}(a)$, and $\mathbb{P}_{\pi_t}(A_t = a) = 0$ otherwise.*

Consider the case where both arm distributions are known to be Bernoulli, such that the only nuisance $\boldsymbol{\eta}$ is $\mu_2$, the mean of arm 2. When $\mu_1^* = \mu_2^* = 1/2$, the limiting distribution of the sample mean test statistic $\rho(H_T)$ is given in Equation 3, where $Z_i$'s denote i.i.d. standard normal variables.

$$\lim_{T \to \infty} \sqrt{T}\left(\rho(H_T) - \mu_1^*\right) \to_d \frac{Z_1 + \mathbf{1}[Z_1 \geq Z_2]\sqrt{2}Z_3}{1 + 2\left(\mathbf{1}[Z_1 \geq Z_2]\right)} \tag{3}$$

If the distribution of observed statistic $\rho(H_T)$ and the simulation-based statistic $\rho(H_T^{(i)})$ are asymptotically equivalent under the correctly specified null $\theta_0 = 1/2$, the CDF of $\rho(H_T^{(i)})$ converges to that of $\rho(H_T)$. This directly implies that Algorithm 2 provides asymptotic control of type I error.

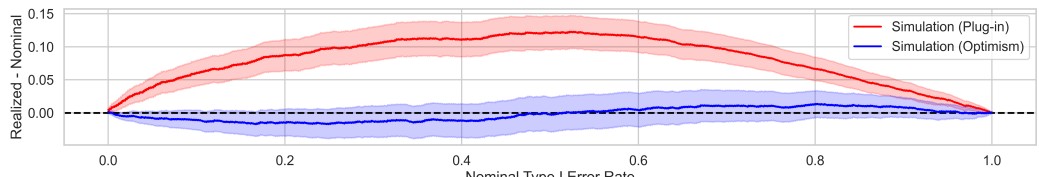

Figure 1: Plot of Realized Type I Error Rate with Plug-in and Optimistic Nuisances for ETC adaptive design with two arms. $Y$-axis corresponds to the difference between realized and nominal (i.e. desired) type I error rates. Shaded regions denote 90% confidence intervals for error rates over 10,000 simulations. Additional details for this set-up are provided in Appendix C.

However, standard estimates for the nuisance $\mu_2$ fail to provide distributional convergence. Using the sample mean estimate $\hat{\mu}_T(2)$, the limiting distribution of $\rho(H_T^{(i)})$ is given by

$$\lim_{T \to \infty} \lim_{B \to \infty} \sqrt{T}\left(\rho(H_T^{(i)}) - \mu_1^*\right) \to_d \frac{Z_1 + \mathbf{1}\left[Z_1 \geq Z_2 + \frac{Z_2 + \mathbf{1}[Z_2 > Z_1]\sqrt{2}Z_4}{1 + 2(\mathbf{1}[Z_2 > Z_1])}\right]\sqrt{2}Z_3}{1 + 2\left(\mathbf{1}\left[Z_1 \geq Z_2 + \frac{Z_2 + \mathbf{1}[Z_2 > Z_1]\sqrt{2}Z_4}{1 + 2(\mathbf{1}[Z_2 > Z_1])}\right]\right)}, \qquad (4)$$

which does not match the distribution of the observed test statistic $\rho(H_T)$.

**Remark 3** (Failure Even With Parametric Rates). *For Example 1, we show in Appendix D that for $\rho(H_T)$ and $\rho(H_T^{(i)})$ to share the same distribution, we require $\sqrt{T}\left(\hat{\mu}_2 - \mu_2^*\right) \to_p 0$. Even with $T$ independent samples from arm 2, the right-hand side of Equation 4 becomes $\frac{Z_1 + \mathbf{1}[Z_1 \geq Z_2 + Z_4/2]\sqrt{2}Z_3}{1 + 2(\mathbf{1}[Z_1 \geq Z_2 + Z_4/2])}$, failing to match the distribution of the observed test statistic $\rho(H_T)$. As a result, asymptotic equivalence in the distribution for the test statistic $\rho(H_T)$ and the simulated distribution of $\rho(H_T^{(i)})$ is unachievable, even if nuisances are estimated at parametric rates on the order of $\Theta\left(1/\sqrt{T}\right)$.*

Given that the distribution of the simulated test statistic $\rho(H_T^{(i)})$ does not converge to that of $\rho(H_T)$, one may wish to avoid nuisance estimation altogether. A simple way to do so is to scan over all possible values of the nuisances that affect both the experiment design and the distribution of the test statistic. In the ETC example above, this means sweeping over all possible values of $\mu_2$ to test a single-point null $\theta = \theta_0$. We reject the null $\theta = \theta_0$ if we reject this null for every value of $\mu_2$ using Algorithm 2. With a grid fidelity of $G$ over arm mean values, $K$ arms, and $B$ number of simulations, we require $G^{K-1}B$ number of experiment simulations to test a single point null $\theta_0$. This approach becomes computationally infeasible rapidly for even moderate values of $K$ and $G$.

### 3.2.1 Preserving Type I Error with Optimistic Nuisances

To avoid such an approach while preserving type I error, we add a small, positive bias to the estimated arm means in the nuisance vector when simulating new trajectories, which we call *simulation with optimism*. By adding positive bias to the mean of arm 2 in our ETC example, we obtain valid type I error control as the number of simulations $B$ grows large. Figure 1 plots the difference in realized and nominal error rates under $\mu_2 = \hat{\mu}_T(2)$ (plug-in) and our optimistic simulation procedure $\mu_2 = \hat{\mu}_T(2) + \log\log N_T(2)/\sqrt{N_T(2)}$. Optimistic nuisances result in type I error rates matching the nominal level, while the plug-in approach results in error rates up to 12% larger than desired.

Beyond our simple ETC example, the principle of optimism when simulating experiment runs controls type I error for a various designs, including reward-maximizing designs that violate positivity conditions necessary for popular reweighing approaches [13] (Example 2) and clipped experimental designs commonly used in practice (Example 3).

**Example 2** (UCB). *Let the arm distributions be any set of distributions $\{P_a\}_{a \in [K]}$ with means $\boldsymbol{\mu}$ such that for all $a \in [K]$, there exists a $B$ such that $P_a$ is $B$-subgaussian. Let $\pi_t(H_{t-1})$ select the arm $A_t = \text{argmax}_{a \in [K]}\left(\hat{\mu}_{t-1}(a) + \sqrt{2\log(T)/N_{t-1(a)}}\right)$.*

**Example 3** (Clipped Reward Maximizing Scheme). *Let the arm distributions be any set of any set of distributions $\{P_a\}_{a \in [K]}$ with finite variance and means $\boldsymbol{\mu}$, where there exists a unique optimal arm $a^*$ such that $\mu_{a^*} > \mu_a$ for all $a \in [K] \setminus \{a^*\}$. Let $\pi_t(H_{t-1})$ be a randomized algorithm such that with probability $\gamma > 0$, we select over the arms uniformly, and with probability $1 - \gamma$, we select an arm such that $\mathbb{P}(A_t \neq a^*) \leq c/t$, where $c$ is a constant independent of $t$.*

We present our results for optimistic nuisance estimation across all examples in Theorem 1, which provides an approach for estimating the nuisance vector $\boldsymbol{\eta}$ for Algorithm 1 that preserves type I error.

**Theorem 1** (Error Control with Optimism). *For nuisance estimation in Algorithm 1, set $\hat{\mu}_a = \hat{\mu}_T(a) + \epsilon_a$, where (i) $\epsilon_a > 0$, and (ii) $\sqrt{\frac{\log \log N_T(a)}{N_T(a)}}/\epsilon_a \to 0$ as $T \to \infty$. Let $\hat{\sigma}_a^2 = \frac{1}{N_T(a)} \sum_{i=1}^{T} \mathbf{1}[A_t = a](X_i - \hat{\mu}_T(a))^2$. Denote Algorithm 2's decision to accept/reject the null hypothesis $\theta_0$ as $\xi(\theta_0, H_T, \alpha)$. Then, for all $\alpha \in [0, 1]$, type I error is asymptotically controlled under Examples 1, 2, and 3 as the number of simulations $B$ grows large, i.e.,*

$$\limsup_{T \to \infty} \lim_{B \to \infty} \mathbb{P}\left(\xi(\theta^*, \alpha, H_T) = 1\right) \leq \alpha. \tag{5}$$

Because we add positive noise to all other mean arms (other than target arm 1), we call our procedure *simulation with optimism*. Intuitively, our approach is inspired by the fact that in many common designs, larger values for other arms' means leads to less samples being allocated to target arm 1. Although fewer samples naturally lead to larger upper (and smaller lower) quantiles for $\rho(H_T)$'s distribution in the i.i.d. case, it is unclear whether this holds for adaptively collected data. In Appendix D, we show that this holds across all examples discussed in Theorem 1.

Among our three examples, Example 1 is unique in that it does not permit asymptotic inference using a stability condition [18] such as Example 2 or with inverse-propensity-weighted estimates such as Example 3. Our approach uniquely addresses designs such as ETC, where (i) the number of arm pulls does not converge in probability as $T \to \infty$ and (ii) observations cannot be reweighted for a similar form of stabilization as in Hadad et al. [13]. Our approach offers a valid form of inference beyond using only the data collected in the exploration period or conservative finite-sample inference.

**Remark 4** (Applicability of Theorem 1). *While Examples 1 and 2 describe explicit sampling schemes (explore-then-commit and one particular version of UCB respectively), Example 3 consists of a broad range of experimental designs. In particular, Example 3 corresponds to regret-optimal (i.e. reward-maximizing) algorithms modified with a positive lower bound on its selection probabilities, provided that the experiment has a unique best arm. To connect regret optimality with the conditions of Example 3, note that the minimum possible regret scales on the order of $\log(T)$ as $T \to \infty$ [8]. As such, a bandit algorithm is only regret optimal if and only if there exists a fixed constant $c$ such that suboptimal arms are pulled at the rate $c/t$ at each timepoint due to $\sum_{t=1}^{T} 1/t \approx \log(T)$. Example 3 therefore captures any reward-maximizing scheme known to be theoretically optimal (e.g. UCB variants such as MOSS-UCB [9], Thompson Sampling [12]) modified with clipping for arm instances with a unique best arm. We note that these clipped schemes are often used for empirical studies in the literature, with applications from political science surveys [23] to digital health interventions [12], where one typically assumes that a best arm exists. Thus, while our examples may seem limited, many real-world applications use the experiment designs covered in Examples 1, 2, and 3.*

**Decay of Bias Term.** The order of the bias $\epsilon$ in Theorem 1 is a direct consequence of the law of iterated logarithm, which states that the difference between the sample mean and the true mean is on the order of $\sqrt{\log \log N_T(a)/N_T(a)}$. By adding positive bias that dominates this term as $T \to \infty$, Theorem 1 ensures that our approach adds asymptotically positive bias to the arm mean nuisances.

**Remark 5** (Selecting bias term $\epsilon$.). *While Theorem 1 provides a lower bound on the order of the bias $\epsilon$, it does not specify an upper bound. In Appendix D, we justify our choice to make $\epsilon \to 0$ as $N_T(a) \to \infty$ using our ETC setup in Example 1. We show that with vanishing bias $\epsilon$, our simulation procedure has higher power, leading to larger probabilities for rejecting mispecified nulls and smaller confidence interval widths. We empirically validate our choice of vanishing bias $\epsilon = \log \log N_T(a)/\sqrt{N_T(a)}$ used in Figure 1 in Appendix C.*

Importantly, simulation with optimistic nuisances preserves the computational tractability of this procedure. While sweeping over all possible nuisance values requires $G^{K-1}B$ number of simulations for type I error control, our hypothesis testing procedure with optimism reduces the number of simulations to simply $B$, removing the runtime dependence on grid fidelity $G$ and number of arms $K$.

---
**Algorithm 3** Confidence Interval and Point Estimate
---
1: **input**: observed data $H_T$, sampling scheme $\pi$, Type I error $\alpha$, simulation number $B$, grid of target parameter values $\Theta_0$.
2: Initialize $\hat{C}(\alpha) = \{\rho(H_T)\}$ as the set containing the sample mean test statistic.
3: **for** $\theta_0$ in $\Theta_0$ **do**
4:    Run Algorithm 2 with null hypothesis $\theta_0$, denoting accept/reject and the estimated quantile value as $\left(\xi(\theta_0, H_T, \alpha), \hat{F}(\theta_0)\right)$.
5:    If $\xi(\theta_0, H_T, \alpha) = 0$, then set $\hat{C}(\alpha) = \hat{C}(\alpha) \cup \{\theta\}$.
6: **Return** Confidence interval $\hat{C}(\alpha)$ and point estimate $\hat{\theta} = \text{argmin}_{\theta \in \hat{C}(\alpha)} |\hat{F}(\theta) - 1/2|$.
---

## 3.3 Confidence Intervals and Point Estimation

Our approach for confidence interval construction and point estimation (provided in Algorithm 3) directly leverages the point null testing approach of Algorithm 2. Given a set of null values, we run our hypothesis testing procedure in Algorithm 2 for each null value. Our confidence interval $\hat{C}(\alpha)$ is the subset of nulls that are not rejected by our hypothesis testing procedure. As our point estimate, we select the null that maximizes the minimum difference between quantile of observed test statistic and quantile rejection thresholds $\{\alpha/2, 1 - \alpha/2\}$, which gives the expression in Algorithm 3. We provide minimal convergence results for our confidence intervals and point estimate in Lemma 2.

**Lemma 2** (Convergence of Point Estimate and Confidence Sequence). *Let Assumption 1 hold, and assume arm variances are finite. Furthermore, assume $\theta^* \in \Theta_0$, i.e. the true null value is contained in the grid of tested nulls in Algorithm 3. Then, for any $\alpha \in [0, 1]$, our confidence interval converges almost surely to a zero-width set containing only $\theta^*$, and our point estimate $\hat{\theta}$ converges to $\theta^*$ almost surely, i.e.*

$$\lim_{T \to \infty} \lim_{B \to \infty} \hat{C}(\alpha) \to_{a.s.} \{\theta^*\}, \quad \lim_{T \to \infty} \lim_{B \to \infty} \hat{\theta} \to_{a.s} \theta^*. \tag{6}$$

Both convergence results in Lemma 2 are direct consequences of Lemma 1, which provides uniform guarantees of rejection for all nulls $\theta_0 \neq \theta^*$. Because all $\theta_0 \neq \theta^*$ are rejected almost surely, the confidence set $\hat{C}(\alpha)$, which consists of nulls that fail to be rejected, converges to $\{\theta^*\}$. Similarly, because $\hat{\theta} \in \hat{C}(\alpha)$, $\hat{\theta}$ must also converge to $\theta^*$.

**Remark 6.** *One may ask whether our assumption that the ground truth value $\theta^*$ lies in the set of tested target parameter values $\Theta_0$ is reasonable, particularly for continuously valued $\theta$. In cases where $\theta$ is known to lie in a bounded interval $\tilde{\Theta}$, one may set $\Theta_0$ to be a grid over $\tilde{\Theta}$ as an approximation. We demonstrate that this does not affect empirical performance in Section 4 below. For unbounded parameters, one may use another confidence interval to get a bounded interval, then use a finely spaced grid over this interval for $\Theta_0$ using an adjusted $\alpha$-level[2].*

**Runtime Relative to Other Methods**  Our runtime results in Appendix C show that to construct our confidence intervals run in reasonable times for moderate values of $G$, the size of the grid, and $B$, the number of simulations per null value. Even when run locally, our confidence intervals take less than a minute to construct. In applications such as political science (shown in our case study example) or healthcare, where data collection is expensive/time-consuming and tight inference is of paramount importance, we expect that runtime of our approach will not be the bottleneck for data analysis. Furthermore, we note that existing approaches often use simulation as a subroutine for their estimation procedure. As discussed in Remark 1, reweighing/MCLT approaches for inference on adaptively collected data require knowledge of the true conditional propensity scores. To obtain the true conditional propensity scores under popular sampling schemes such as Thompson sampling (which does not have clear, closed-form propensity scores such as $\epsilon$-greedy sampling schemes), works such as [36] and [13] simulate the sampling policy $\pi$ using history $H_{t-1}$ to obtain conditional propensity scores $\pi_t$ at each time point. While the reweighing/MCLT approaches do not directly require resimulation, it is common practice to estimate the true propensity scores needed for reweighing approaches by simulating the known sampling policy.

---

[2]We discuss this approach in Appendix A using the anytime valid bounds of Waudby-Smith et al. [32]

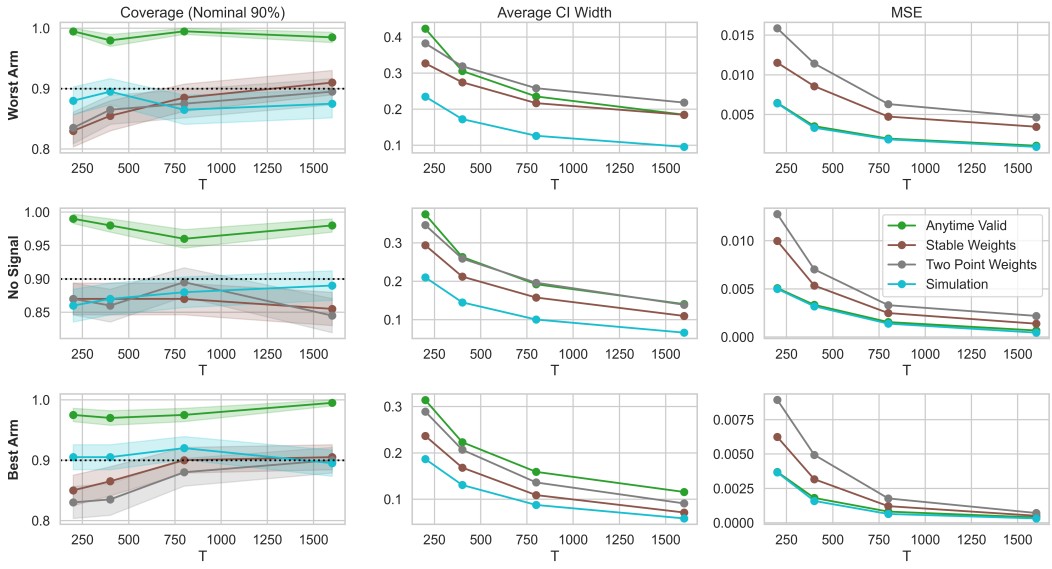

Figure 2: Coverage probabilities, average CI widths, and MSE for synthetic setup, with $T$ values 200, 400, 800, 1600. Results are averaged over 200 simulations. Shaded region denotes 1 standard error.

## 4 Experimental Results

We provide experimental results for type I error and confidence interval widths across both synthetic and real-world data. For all experiments, we set type I error rate $\alpha = 0.1$, and set $\epsilon = \log\log N_T(a)/\sqrt{N_T(a)}$, which satisfies the conditions of Theorem 1. We provide additional details, including runtime, baseline pseudocode, and results for alternative setups, in Appendix C.

**Synthetic Experiment Setup.** For the synthetic experiments, we set $K = 3$ and set our target parameter to be the mean of arm 1. All arms are distributed according to a Bernoulli distribution with mean vectors $\boldsymbol{\mu} = [0.45, 0.5, 0.55]$, $[0.5, 0.5, 0.5]$, and $[0.55, 0.5, 0.45]$ corresponding to the worst arm, no signal, and best arm settings respectively. For our confidence intervals, we test a grid of 100 null values evenly spaced between $[0, 1]$, the range of mean values, with $B = 200$ simulations per mean value. To compute mean square error (MSE), we use the null $\theta_0$ with $p$-value $\hat{F}(\rho(H_T))$ closest to $0.5$ as a heuristic point estimate. As our baselines, we test three distinct approaches for valid inference for adaptively collected data: (i) an empirical Bernstein anytime valid approach [31], (ii) an asymptotic approach based on variance stabilizing weights [13, 36], and (iii) an asymptotic approach based on variance minimizing weights [13]. To accommodate the latter two approaches, we present results using a modified UCB (Example 2) scheme with clipping at the rate of $t^{-0.7}$.

**Real-World Data.** To assess the performance of our approach on real-world data, we reanalyze the results of an adaptive experiment run by Offer-Westort et al. [23]. The adaptive experiment design used a batched Thompson sampling procedure with $T = 1000$ on Amazon's MTurk platform. Arms correspond to different wordings of the ballot measure, and outcomes corresponding to binary responses indicating whether the respondent would support the measure. As an additional baseline, we provide the intervals reported by Offer-Westort et al. [23]. For each confidence interval based on our simulation procedure, we test a grid of 200 null values evenly spaced between $[0, 1]$, with $B = 1000$ simulations per null value, and use the same heuristic as above for our point estimate.

The sampling policy for the collected data falls under Example 3. In Appendix B.1. of Offer-Westort et al. [23], the authors state "For estimation and hypothesis testing, we have assumed that there is a unique best arm." Empirical results in Figures 3 and 4 of [23] (right-side figures correspond to our data) suggest that this is true. In page 19 of [23], the authors discuss clipping, which was enforced through sampling with Thompson sampling with probability 0.9, and uniformly at random with probability 0.1. This satisfies the clipping requirement, where $\gamma = 0.1$ as defined in Example

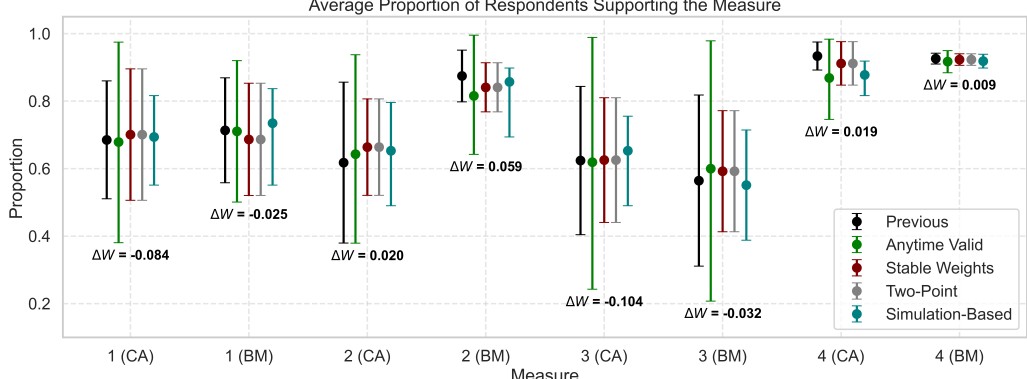

Figure 3: 90% Confidence Intervals for real-world data. The difference in CI widths between our simulation-based approach and the smallest width interval among baselines is annotated as $\Delta W$.

3. Existing work [17] has shown that batched TS preserves the problem-specific regret optimality. Following the argument outlined in Remark 4, it follows that there exists a fixed constant $c < \infty$ such that $\mathbb{P}(A_t \neq a^*) \leq c/t$. Thus, our approach offers valid inference on this dataset.

**Discussion of Results.** Our empirical results demonstrate the key benefits of our simulation-based approach: across both synthetic and real-world data, our simulation-based confidence intervals tend to produce smaller confidence intervals, while maintaining similar coverage (e.g. type I error) as existing approaches. Across all experiments, confidence interval widths are reduced by as much as 50% relative to the next best method, demonstrating the benefits of simulation-based inference.

Figure 2 plots the results of our synthetic experiments with respect to $T$, the total duration of the experiment. The rates of coverage (i.e. the probability that the null $\theta = \mu_1^*$ is not rejected) demonstrates that our approach provides similar type I error guarantees to the two other asymptotic methods. Given similar coverage/error rates, our simulation-based confidence intervals and point estimates provide the tightest confidence intervals on average and smallest MSE across all setups. The largest gains, particularly in terms of confidence interval width, are in the setting where we conduct inference on the worst arm, where we see up to 50% reductions in average width.

Our real-world experiments demonstrate similar results to our synthetic setup. For arms that appear to be the worst-performing (such as Proposal 3 (CA)), our simulation-based approach reduces confidence interval widths drastically. The three proposals with the lowest support see reductions of up to roughly 50% in terms of confidence interval width relative to the intervals reported by Offer-Westort et al. [23]. While the best performing arms see a slight increase in interval width, we note that the width decreases in the worst performing arms are significantly larger than the width gains in the best performing arms. Intervals grow by at most $0.059$ relative to the best performing baseline, while being decreasing widths up to $0.1$. Furthermore, simulation outperforms all existing baselines for a majority of treatment arms and is never widest among all approaches.

## 5   Conclusion

This paper introduces a simulation-based method for conducting inference in experiments with adaptive designs, where traditional confidence intervals and hypothesis tests often fail. By simulating the experiment under the null hypothesis using *optimistic* nuisances with positive bias, the method ensures valid type I error control. Across both synthetic and real-world experiments, empirical results show that our simulation-based approach significantly reduces confidence interval widths—up to 50% smaller for undersampled arms—while maintaining accurate statistical coverage. Future directions of our method include extensions for (i) additional experiment designs beyond our examples, (ii) valid error control under settings with contextual information, (iii) inference approaches that allow sequential testing across time.

## Acknowledgments

We thank the anonymous reviewers for their thoughtful feedback and insightful suggestions, which have helped improve and strengthen this work. Brian M. Cho was supported by the Department of Defense (DoD) through the National Defense Science & Engineering Graduate (NDSEG) Fellowship Program. Nathan Kallus was supported by the U.S. National Science Foundation under Grant No. 1846210. Any opinions, findings, and conclusions or recommendations expressed in this material are those of the author(s) and do not necessarily reflect the views of the U.S. Department of Defense or the U.S. National Science Foundation.

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

# A  Additional Results for Inference on Means

## A.1  Intuition on Vanishing Bias $\epsilon_a$

A natural question is why we choose the bias term to vanish. Under the conditions of Theorem 1, simply setting $\hat{\mu}_a = 1$ (the upper bound of the support) preserves Type I error. To see the value of vanishing positive bias, consider the ETC design discussed in Section 3.2, but $\mu_2^* < \mu_1^* < 1$. The limit distribution of $\rho(H_T)$, the sample mean of arm 1, takes the form:

$$\lim_{T \to \infty} \sqrt{T} \left( \rho(H_T) - \mu_1^* \right) \to_d \frac{2\sigma_1^* Z_1 + 2\sqrt{2}\sigma_1^* Z_2}{3} = \frac{2\sigma_1^* Z_3}{\sqrt{3}}, \tag{7}$$

where $\sigma_1^* = \sqrt{\mu_1^*(1 - \mu_1^*)}$ is the true standard deviation of arm 1 and $Z_1, Z_2, Z_3$ are i.i.d. normal random variables. If we set the nuisance value $\hat{\eta} = \hat{\mu}_2$ equal to the maximum possible value of 1, then, for any null value $\theta_0$, the distribution of the test statistic using simulated trajectories $H_T^{(i)}$ is given by

$$\lim_{T \to \infty} \sqrt{T} \left( \rho(H_T^{(i)}) - \theta_0 \right) \to_d 2Z_1 \sqrt{(1 - \theta_0)\theta_0} \tag{8}$$

for all $\theta_0 \in [0, 1)$. In contrast, under a simulation procedure where $\hat{\eta} \to \mu_2^*$, the limiting distribution of the test statistic takes a piecewise form, given by the following:

$$\lim_{T \to \infty} \sqrt{T} \left( \rho(H_T^{(i)}) - \theta_0 \right) \to_d \frac{2Z_1 \sqrt{(1 - \theta_0)\theta_0} + (\mathbf{1}[\theta_0 > \mu_2]) Z_3 \sqrt{8(1 - \theta_0)\theta_0}}{1 + 2(\mathbf{1}[\theta_0 > \mu_2])}. \tag{9}$$

While the power (i.e. the probability of rejection when $\theta_0 \neq \mu_1^*$) is unchanged for values of $\theta_0 < \mu_2^*$, a vanishing bias term results in improved power for all values of $\theta_0 > \mu_2^*$. When $\theta_0 > \mu_2$, the simulated distribution of the test statistic $\rho(H_T^{(i)})$ is more tightly centered around the value $\theta_0$ compared to the choice of $\hat{\mu}_2 = 1$ by a factor of $1/\sqrt{3}$. As a result, the probability that our observed test statistic $\rho(H_T)$ lies beyond the lower and upper quantiles of simulated distribution is larger under nuisances $\hat{\eta}$ that converge to $\eta^*$.

## A.2  Confidence Intervals for Unbounded Parameter Spaces.

While the main body of our paper focuses on bounded parameter spaces (e.g. $\theta^* \in [0, 1]$), one can construct a bounded region of the parameter space by using the confidence interval in Theorem 2.2 of Waudby-Smith et al. [32] with $\alpha_1$, then use $\alpha_2$ for Algorithm 2, where $\alpha_1 + \alpha_2 = \alpha$. This maintains statistical validity by a union-bound argument. Note that to preserve power as close as possible to our simulation procedure, one should set $\alpha_2 \gg \alpha_1$, as $\alpha_1$ is only used to construct a bounded set over which we construct a fine grid.

# B  Inference on Difference in Means

For our difference-in-means parameter, our approach is almost unchanged. Without loss of generality, assume that our target parameter is now $\theta = \mu_1 - \mu_2$. We can use the same exact approach as the main boyd of our paper, where we use a plug-in estimate $\hat{\mu}_2$ with positive bias and now vary $\theta = \mu_1 - \mu_2$. Note that in effect, varying $\theta = \mu_1 - \mu_2$ is the same as varying $\theta = \mu_1$ in our simulations. We opt for the same test statistic as before.

Note that to preserve statistical validity, one cannot select $\mu_1$ (or $\mu_2$) into the nuisance vector $\hat{\eta}$ while looking at the data. However, if one knows that the design will pull an arm more often (e.g. control-augmented sampling such as Offer-Westort et al. [23]), then one should use the arm that will be pulled more often to improve power.

One may ask why we do not recommend using a test statistic such as the difference in sample means. Note that by using two sums (rather than one) as our test statistic, the variance of our test statistic increases, and therefore results in less power. As such, randomly selecting one of the arms' sample mean as the test statistic for the difference-in-means target parameter will have higher power (lower variance) than including both means.

# C Additional Experimental Results

All computational results in both the main body and the appendix were run locally on a 14-inch MacBook Pro with an Apple M2 Pro chip and 16GB of memory.

## C.1 Runtime

Our confidence interval runtimes (i.e. the time to construct a confidence set) depend on $G$, the number of point nulls in $\Theta_0$, our set of nulls, and $B$, the number of simulations done per null. In particular, our runtime scales on the order of $O(GB)$. To demonstrate this scaling, we provide runtime results in Table 1, using the real-world dataset collected under a batched Thompson sampling scheme.

| $G$ | $B$ | Runtime (seconds) |
|-----|-----|-------------------|
| 50  | 50  | $11.50 \pm 0.22$  |
| 50  | 100 | $20.21 \pm 0.32$  |
| 100 | 50  | $20.66 \pm 0.30$  |
| 100 | 100 | $39.17 \pm 0.36$  |

Table 1: Table of runtimes for constructing confidence intervals for the real-world dataset.

Our table verifies our runtime scaling. As $G$ doubles, the runtimes doubles exactly. Likewise, as $B$ doubles, the runtime also doubles. We note that these runtimes are obtained by running this code locally. Running experiments on a cluster or a HPC setup will likely improve performance, but runtimes are reasonable even even on local devices. As such, we do not believe our approach is prohibitively expensive computationally.

## C.2 Additional Experiment Details

**Synthetic Experiments** We provide additional pseudocode for all sampling schemes used in our experiments. We begin with our clipped modifications to UCB, where we provide our clipping procedure in Algorithm 4.

---
**Algorithm 4** Clipped Adaptive Design with Decaying Exploration
---
1: **Input**: horizon $T$, arm number $K$, sampling scheme $\pi$, decay rate $\beta \in [0, 1]$.
2: Initialize $\hat{\mu}_0(a) = 0$, $N_T(a) = 0$ for all $a \in [K]$.
3: Sample each arm $a \in [K]$ once.
4: **for** $t \in [T]$ **do**
5:     Sample $b \sim \text{Unif}[0, 1]$.
6:     **if** $b \leq t^{-\beta}$ **then**
7:         Sample an arm $A_t \in [K]$, with $\mathbb{P}(A_t = a | H_{t-1}) = 1/K$.
8:     **else**
9:         Sample an arm $A_t \sim \pi_t(H_{t-1})$.
10: **Return** trajectory $H_T$.

---

For our clipped UCB scheme, we use the sampling scheme $\pi$ described in Example 2. Another choice of $\pi$ we test below is the $\epsilon$-greedy scheme, where $\pi_t(H_{t-1})$ selects the arm $A_t = \text{argmax}_{a \in [K]} \hat{\mu}_{t-1}(a)$, and explore based on the decay rate $\beta \in [0, 1]$. For both our clipped UCB and $\epsilon$-greedy scheme, we set $\beta = 0.7$, matching the exploration decay rates found in Hadad et al. [13].

**Real-World Experiments** The real-world dataset we use for constructing confidence intervals was collected by Offer-Westort et al. [23] on the Amazon MTurk Platform. Here, 1000 participants were recruited from June 21st, 2018 to June 30th, 2018, where each subject was paid 1\$ for their participation. We construct confidence intervals on the RTW treatments and responses. The wordings of each ballot measure can be found in Table 2 of Offer-Westort et al. [23], and are based on ballot measures in Missouri, North Dakota, Oklahoma, and South Dakota.

**Details for Figure 1**    In Figure 1, we generate observations from two arms, both with rewards $X_t \sim N(0, 1)$. We slightly modify our ETC example, where we sample both arms $T/10$ times, and commit to the arm with the largest empirical mean at time $T/5$ for the remaining duration of $4T/5$. We set $T = 5000$, with 10,000 replications for Figure 1.

### C.3    Additional Experiment Results

We provide additional experimental results regarding (i) a different sampling scheme as our adaptive design and (ii) a different choice of bias for the ETC example in Figure 1.

**$\epsilon$-Greedy Design**    We choose the $\epsilon$-greedy design as an additional candidate for our approach. This design is known to have poor limiting distribution properties (see the first figure of Khamaru and Zhang [18]). We present our results in Figure 5. Our conclusions remain unchanged, as we see the largest gains (e.g. decreases in interval width) for the means of arms that are sampled less often. Interestingly, we see that our heuristic for selecting the point estimate outperforms all other approaches here. We plan to investigate this in future work by providing theoretical guarantees regarding our point estimate approach.

**Choice of Bias Term**    To investigate our choice of optimistic bias term, we test multiple different choices of $\epsilon_a$ in Figure 4 under the same setting as Figure 1.

- Bias 1 denotes $\epsilon_a = \frac{\log \log N_T(a)}{\sqrt{N_T(a)}}$.

- Bias 2 denotes $\epsilon_a = \frac{\log N_T(a)}{\sqrt{N_T(a)}}$

- Bias 3 denotes $\epsilon_a = 1$.

Bias 2 and 3 are intended to demonstrate that our choice of bias term (Bias 1), which is close to the lower bound rate of $\sqrt{\log \log N_T(a)/N_T(a)}$ in Theorem 1, is preferable in practice. While All three methods provide appropriate type I error control (as defined in Definition 1), Bias 1 (the bias term used in our paper) demonstrates superior power for nulls $\theta_0 = 0.02$ close to the ground truth value of $\theta^* = 0$. This suggests that the bias term $\epsilon_a$ should be set as close to $\sqrt{\log \log N_T(a)/N_T(a)}$ as possible.

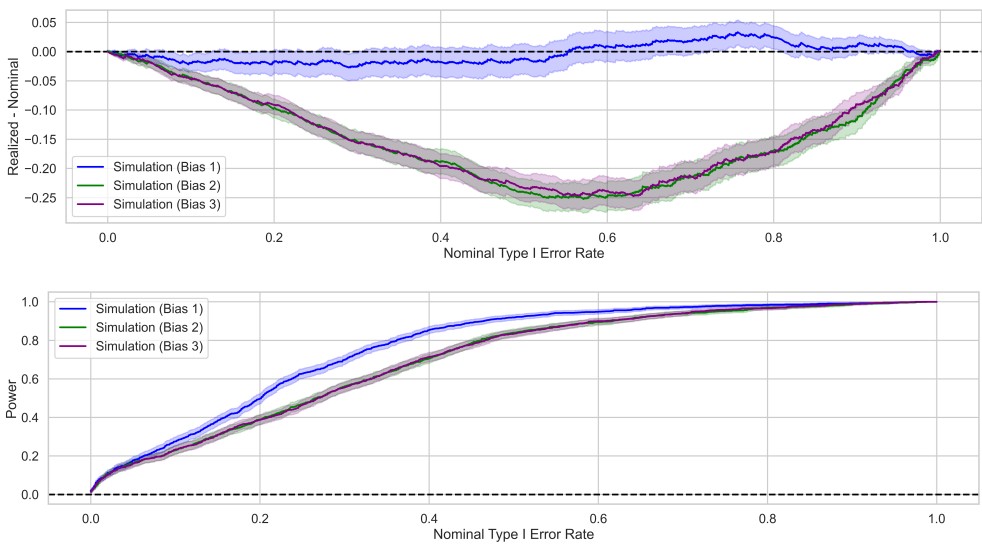

Figure 4: Top: Type I error rates based on magnitude of bias term $\epsilon_a$, using the null $\theta^* = 0$. Bottom: Power against the null $\theta_0 = 0.02$

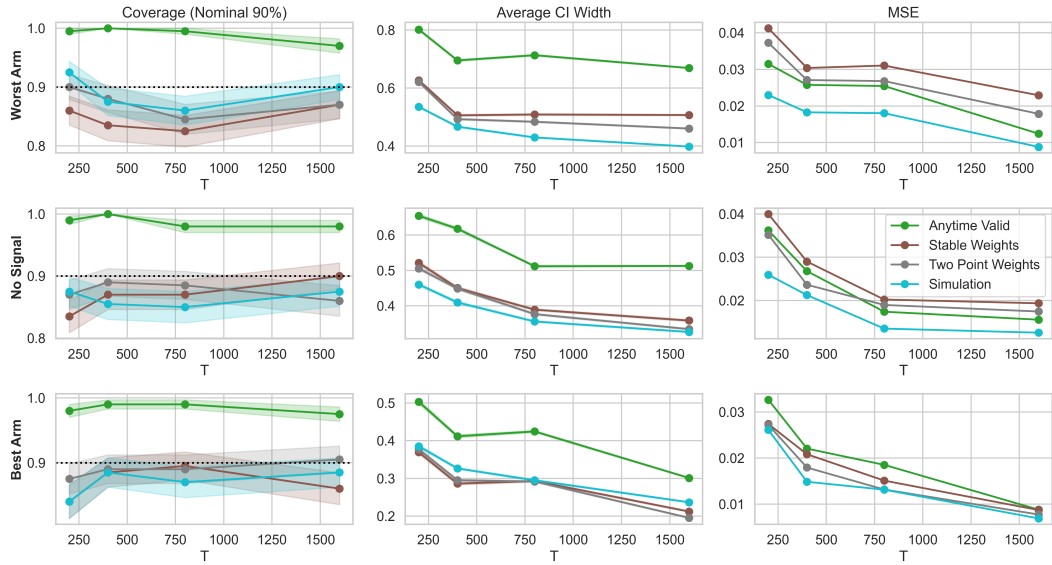

Figure 5: Coverage probabilities, average CI widths, and MSE for synthetic setup using the clipped $\epsilon$-greedy scheme, with $T$ values 200, 400, 800, 1600. Results are averaged over 200 simulations. Shaded region denotes 1 standard error.

**Violation of Assumptions** To test whether our approach is sensitive to violations of our assumptions, we test Beta-Bernoulli Thompson sampling *without* batching and clipping, a design that does not satisfy the requirements of any of our examples. We provide the results for inference on the best arm and the worst arm for a three-armed experiment with $\mu = [0.55, 0.5, 0.45]$. We set $\alpha = 0.1$ and $T = 400$, and all other parameters identical to those used in Section 4. We note that we set the horizon to a relatively small number to avoid issues with zero propensity scores (which would make our baselines produce trivial results). We report our results with 100 simulations of our method in the Tables 2 and 3 below, with one standard deviation provided in the $\pm$ terms.

| Method | Coverage | CI Width | MSE |
|---|---|---|---|
| Anytime-Valid | $0.98 \pm 0.01$ | $0.23 \pm 0.002$ | $0.006 \pm 0.000$ |
| Stable Weights | $0.89 \pm 0.03$ | $0.14 \pm 0.002$ | $0.004 \pm 0.000$ |
| Two Point Weights | $0.84 \pm 0.04$ | $0.16 \pm 0.002$ | $0.006 \pm 0.000$ |
| Simulation | $0.87 \pm 0.03$ | $0.13 \pm 0.002$ | $0.004 \pm 0.000$ |

Table 2: Inference on best arm mean $\mu_3 = 0.55$.

| Method | Coverage | CI Width | MSE |
|---|---|---|---|
| Anytime-Valid | $0.88 \pm 0.03$ | $0.31 \pm 0.001$ | $0.005 \pm 0.000$ |
| Stable Weights | $0.69 \pm 0.05$ | $0.18 \pm 0.001$ | $0.005 \pm 0.000$ |
| Two Point Weights | $0.69 \pm 0.05$ | $0.21 \pm 0.001$ | $0.006 \pm 0.000$ |
| Simulation | $0.88 \pm 0.03$ | $0.21 \pm 0.001$ | $0.004 \pm 0.000$ |

Table 3: Inference on worst arm mean $\mu_1 = 0.45$.

Our results demonstrate that relative to the baselines, the simulation procedure provides tighter confidence intervals without sacrificing type I error. When conducting inference on the best arm, our approach achieves close to the nominal coverage level, while producing the tightest confidence interval among all methods. For the worst arm, we see that stable weights and two point weights, inference approaches based on martingale central limit theorems (MCLT), provide relatively small confidence intervals, but suffer from significant undercoverage. In contrast, both our anytime-valid baseline and our simulation approach result in coverage close to nominal levels. Between these two methods, our simulation approach produces much smaller confidence intervals, demonstrating its

benefits empirically. Thus, while we may currently lack theoretical guarantees for general sampling algorithms, our simulation approach retains its competitive inference performance without sacrificing type I error empirically.

# D   Proofs of Theoretical Results

We use $\omega$ to index sample paths from the set of all sample paths $\Omega$, where $P(\Omega) = 1$. For a random variable $X$, we use the notation $X(\omega)$ to index its sample path. Before providing our proofs, we provide a technical lemma from existing work that provides almost sure convergence guarantees. For completeness, we provide these results below.

**Lemma 3** (Fact E.1, Shin et al. [27]). *Suppose that there exists a process $(Y_n)_{n=1}^{\infty}$ indexed by n, and $Y_n \to Y$ a.s. as $n \to \infty$. Furthermore, assume that there exists an index process $(N(t))_{t=1}^{\infty}$ indexed by t such that $N(t) \to \infty$ a.s. as $t \to \infty$. Then $Y_{N(t)} \to Y$ as $t \to \infty$.*

We also provide a simplified version of the law of iterated logarithm, which underlies the magnitude of the positive bias term $\epsilon_a$ added to preserve type I error.

**Lemma 4** (Law of Iterated Logarithm [32]). *Let $(X_i)_{i=1}^{\infty}$ be a sequence of i.i.d. random variables with mean $\mu$ and variance $\sigma^2 < \infty$. Let $\hat{\mu}_T = \frac{1}{t} \sum_{i=1}^{T} X_i$. Then, the following holds almost surely:*

$$\limsup_{T \to \infty} \frac{\sqrt{T}|\hat{\mu}_T - \mu|}{\sigma\sqrt{2 \log \log T}} = 1. \tag{10}$$

As a direct application of Lemmas 3 and 4, we obtain a simple, yet useful result regarding our optimistic nuisances $\hat{\mu}_a$.

**Corollary 1** (Almost-Sure Positive Bias). *Let $\hat{\mu}_a = \hat{\mu}_T(a) + \epsilon_a$, where $\hat{\mu}_T(a)$ denotes the sample mean up to time $t$ and $\epsilon_a$ satisfies the conditions posed in Theorem 1. Let Assumption 1 hold, and assume arm variances are finite. Then, as $T \to \infty$, $\hat{\mu}_a \geq \mu_a^*$ almost surely, i.e.*

$$\mathbb{P}\left(\limsup_{T \to \infty}\{\omega \in \Omega : \mathbf{1}\left[\hat{\mu}_a(\omega) \geq \mu_a^*\right] = 0\}\right) = 0. \tag{11}$$

*Proof of Corollary 1.* This follows directly from Lemmas 3 and 4. For completeness, we provie this result by contradiction. Assume there exists a set of sample paths $\Omega' \subseteq \Omega$ such that $\mathbb{P}(\omega') > 0$, and $\mathbb{P}(\limsup_{T \to \infty}\{\omega \in \Omega' : \mathbf{1}[\hat{\mu}_a(\omega) \geq \mu_a^*] = 0\}) = 1$. Then, this implies that for $\omega \in \Omega'$, the following must hold:

$$\limsup_{\omega \in \Omega', \, T \to \infty} \hat{\mu}_a(\omega) - \mu_a^* = \limsup_{\omega \in \Omega', \, T \to \infty} \hat{\mu}_T(a)(\omega) + \epsilon_a(\omega) - \mu_a \leq 0. \tag{12}$$

Multiplying by $\frac{\sqrt{N_T(a)(\omega)}}{\sigma_a^*\sqrt{2 \log \log N_T(a)(\omega)}}$ on both sides, we obtain

$$\limsup_{\omega \in \Omega', \, T \to \infty} \frac{\sqrt{N_T(a)(\omega)}\,(\hat{\mu}_T(a)(\omega) - \mu_a^*)}{\sigma_a^*\sqrt{2 \log \log N_T(a)(\omega)}} + \frac{\sqrt{N_T(a)(\omega)}}{\sigma_a^*\sqrt{2 \log \log N_T(a)(\omega)}}\epsilon_a \leq 0. \tag{13}$$

By the condition on $\epsilon_a$ in Theorem 1, the second term on the LHS diverges to infinity, so the first term must diverge to negative infinity. However, note that by Lemmas 3 and 4, the following must hold:

$$\limsup_{\omega \in \Omega, T \to \infty} \frac{\sqrt{N_T(a)(\omega)}|\hat{\mu}_T(a)(\omega) - \mu_a^*|}{\sqrt{2 \log \log N_T(a)(\omega)}} = 1. \tag{14}$$

Because $\Omega' \subseteq \Omega$, we obtain the following inequality:

$$\limsup_{\omega \in \Omega', T \to \infty} \frac{\sqrt{N_T(a)(\omega)}|\hat{\mu}_T(a)(\omega) - \mu_a^*|}{\sqrt{2 \log \log N_T(a)(\omega)}} \leq \limsup_{\omega \in \Omega, T \to \infty} \frac{\sqrt{N_T(a)(\omega)}|\hat{\mu}_T(a)(\omega) - \mu_a^*|}{\sqrt{2 \log \log N_T(a)(\omega)}} = 1, \tag{15}$$

which results in a contradiction and therefore completes our proof. $\square$

Lastly, we present a known result regarding asymptotic normality of sample means.

**Lemma 5** (Asymptotitc Normality under Stability [18]). *Let $P_a$ have finite variance, and assume that there exists a constant $N_a^*(T)$ such that the following holds:*

*(i) $N_T(a) \to_P N_a^*(T)$ as $T \to \infty$*

*(i) $N_a^*(T) \to \infty$ as $T \to \infty$.*

*Then, whenever $\hat{\sigma}_a$ is a consistent estimate of $\sigma_a^*$, $\sqrt{\frac{N_T(a)}{\hat{\sigma}_a}} (\hat{\mu}_T(a) - \mu_a^*) \to_d N(0,1)$.*

We will leverage this result heavily for the proof of Examples 2 and 3 in Theorem 1.

### D.1 Proof of Lemma 1

By the infinite sampling condition in Assumption 1, as $T \to \infty$, $N_T(a) \to \infty$ for all $a \in [K]$ for any underlying arm distributions $\{P_a\}_{a \in [K]}$. Note that this includes any potential choice of null $\theta_0$ used in Algorithms 1 and 2. By direct application of Lemma 3 and the strong law of large numbers (SLLN), under any choice of null value $\theta_0$, the sample mean test statistic $\rho(H_T^{(i)}) \to_{a.s.} \theta_0$. By definition of almost sure convergence, using $i$ to denote the simulation number and $\omega$ to index the sample path of the observed experiment,

$$\mathbb{P}\left(\limsup_{T \to \infty}\{i \in [B], \omega \in \Omega : |\rho(H_T^{(i)})(\omega) - \theta_0| > \epsilon\}\right) = 0 \quad \text{for all } \epsilon > 0. \tag{16}$$

For any $\theta_0 \neq \theta^*$, let $\epsilon^* = \frac{|\theta_0 - \theta^*|}{3}$. As $T \to \infty$, for all $\omega \in \Omega$, Equation 16 implies that there exists an $T_1(\omega) < \infty$ such that $\sup_{i \in [B]} |\rho(H_T^{(i)})(\omega) - \theta_0| < \epsilon^*$ for all $T > T_1(\omega)$.

For the observed test statistic $\rho(H_T)$, by Assumption 1 and Lemma 3, we obtain $\rho(H_T) \to_{a.s.} \theta^*$, i.e. for all $\omega \in \Omega$, there exists a $T_2(\omega) < \infty$ such that $|\rho(H_T)(\omega) - \theta^*| < \epsilon^*$ for all $T > T_2(\omega)$.

Putting these results together, we obtain that for any $\omega \in \Omega$, there exists a finite $T^*(\omega) = \max(T_1(\omega), T_2(\omega)) < \infty$ such that $\sup_{i \in [B]} |\rho(H_T^{(i)}) - \theta_0| < \epsilon^*$ and $|\rho(H_T) - \theta^*| < \epsilon^*$. As a result, we obtain the quantile of $\rho(H_T)$ with respect to the simulated test statistic distribution $\left(\rho(H_t^{(i)})\right)_{i=1}^{B}$ is either zero or one. To see this, without loss of generality, assume that $\theta_0 < \theta^*$. Then, for $T > T^*(\omega)$, denoting $\hat{F}(\cdot)$ as the quantile function on $\left(\rho(H_t^{(i)})\right)_{i=1}^{B}$, for all $\omega \in \Omega$,

$$\hat{F}(\rho(H_T)(\omega)) = \frac{1}{B}\sum_{i=1}^{B} \mathbf{1}[\rho(H_T)(\omega) \leq \rho(H_T^{(i)})(\omega)] \tag{17}$$

$$= \frac{1}{B}\sum_{i=1}^{B} \mathbf{1}\left[(\rho(H_T)(\omega) - \theta^*) - \left(\rho(H_T^{(i)})(\omega) - \theta_0\right) + \theta^* - \theta_0 \leq 0\right] \tag{18}$$

$$\leq \frac{1}{B}\sum_{i=1}^{B} \mathbf{1}\left[-\left(\underbrace{|\rho(H_T)(\omega) - \theta^*|}_{<\epsilon^*} + \underbrace{|\rho(H_T^{(i)})(\omega) - \theta_0|}_{<\epsilon^*}\right) + \theta^* - \theta_0 \leq 0\right] \tag{19}$$

$$\leq \frac{1}{B}\sum_{i=1}^{B} \leq \mathbf{1}\left[-\frac{2(\theta^* - \theta_0)}{3} + (\theta^* - \theta_0) \leq 0\right] \tag{20}$$

$$= \frac{1}{B}\sum_{i=1}^{B} \mathbf{1}\left[\frac{\theta^* - \theta_0}{3} \leq 0\right] = 0. \tag{21}$$

Note that $\hat{F}$ is bounded between 0 and 1, so $\hat{F}(\rho(H_T)(\omega))$ must be 0. Line 19 follows from the almost sure convergence results discussed previously and the fact that the indicator function monotonically increases as the LHS inside the indicator decreases. Line 20 follows by definition from the definition of $\epsilon^*$. Lastly, our final result follows from our assumption that $\theta^* > \theta_0$. Our proof proceeds analogously in the case where $\theta_0 > \theta^*$, with the end result showing that $\hat{F}(\rho(H_T)(\omega)) = 1$.

As a result, we obtain that for any $\omega \in \Omega$, we reject $\theta_0 \neq \theta^*$ almost surely in Algorithm 2, i.e.

$$\mathbb{P}\left(\omega \in \Omega : \lim_{T \to \infty} \hat{F}\left(\rho(H_T)(\omega)\right) \in \{0, 1\}\right) = 1. \tag{22}$$

## D.2 Proof of Theorem 1

We prove the results for Theorem 1 for each case separately, beginning with Example 1. Before proceeding, we prove a result regarding the convergence of the variance estimators in Lemma 6.

**Lemma 6** (Convergence of Variance Estimates.). *Let Assumption 1 hold, let arm variances be finite, and let $\hat{\sigma}_a^2$ be defined as in Theorem 1. Then, our variance estimate is strongly consistent, i.e. $\hat{\sigma}_a^2 \to_{a.s.} \sigma_a^2$.*

*Proof of Lemma 6.* To prove this result, we first expand the variance estimator as follows:

$$\hat{\sigma}_a^2 = \frac{1}{N_T(a)} \sum_{i=1}^{T} \mathbf{1}[A_t = a](X_i - \hat{\mu}_T(a))^2 \tag{23}$$

$$= \underbrace{\frac{1}{N_T(a)} \sum_{i=1}^{T} \mathbf{1}[A_t = a]X_i^2}_{(a)} - \underbrace{\left(\frac{1}{N_T(a)} \sum_{i=1}^{T} \mathbf{1}[A_t = a]X_i\right)^2}_{(b)} \tag{24}$$

We now show that $(a) \to_P (\mu_a^*)^2 + (\sigma_a^*)^2$ and $(b) \to_P (\mu_a^*)^2$.

For term (b), note that by Assumption 1 and Lemma 3, $\left(\frac{1}{N_T(a)} \sum_{i=1}^{T} \mathbf{1}[A_t = a]X_i\right) \to_{a.s.} \mu_a^*$. By the continuous mapping theorem, $(b) \to_{a.s.} (\mu_a^*)^2$.

For term (a), we can further decompose the estimate as follows:

$$(a) = \frac{1}{N_T(a)} \sum_{i=1}^{T} \mathbf{1}[A_t = a](X_i - \mu_a^* + \mu_a^*)^2 \tag{25}$$

$$= \underbrace{\frac{1}{N_T(a)} \sum_{i=1}^{T} \mathbf{1}[A_t = a](X_i - \mu_a^*)^2}_{(i)} + \underbrace{\frac{1}{N_T(a)} \sum_{i=1}^{T} \mathbf{1}[A_t = a]2(X_i - \mu_a^*)(\mu_a^*) + (\mu_a^*)^2}_{(ii)} \tag{26}$$

Term $(i)$ converges almost surely to $(\sigma_a^*)^2$ by definition of variance, the strong law of large numbers, and Lemma 3. Term $(ii)$ converges to 0 almost surely due to the fact that $\frac{1}{N_T(a)} \sum_{i=1}^{T} \mathbf{1}[A_t = a](X_i - \mu_a^*) \to_{a.s.} 0$. As a result, we obtain $(a) \to_{a.s.} (\sigma_a^*)^2 + (\mu_a^*)^2$.

Putting the convergence results of $(a)$ and $(b)$ together, we obtain that $\hat{\sigma}_a^2 \to_{a.s.} (\sigma_a^*)^2$. $\square$

We also provide a simple, but useful result regarding the limiting type I error control under two key conditions.

**Lemma 7** (Asymptotic Type I Error Control). *Let $\omega \in \Omega$ denote a sample path, such that the set of sample paths has probability 1, i.e. $\mathbb{P}(\Omega) = 1$. Let $F(\cdot)$ and $\hat{F}(\cdot)(\omega)$ denote the CDF of the test statistic distribution for observed test statistic $\rho(H_T)$ and simulated test statistic $\rho(H_T^{(i)})$ under sample path dependent nuisances $\hat{\eta}(\omega)$. Let $F^{-1}$ and $\hat{F}^{-1}(\omega)$ denote their respective quantile functions. Assume that the following conditions hold:*

*(i)* $\limsup_{T \to \infty} F\left(\sup_{\omega \in \Omega} \hat{F}^{-1}(\alpha/2)(\omega)\right) + \left(1 - F\left(\inf_{\omega \in \Omega} \hat{F}^{-1}(1 - \alpha/2)(\omega)\right)\right) \leq \alpha$,

*(ii)* $\mathbb{P}\left(\rho(H_T) \in \left\{\hat{F}^{-1}(\alpha/2)(\omega), \hat{F}^{-1}(1 - \alpha/2)(\omega)\right\}\right) = 0$ *for all $\omega \in \Omega$.*

*Then, denoting the decision for rejecting/accepting the ground truth null $\theta^*$ as $\xi(\theta^*, \alpha, H_T)$,*

$$\limsup_{T \to \infty} \lim_{B \to \infty} \mathbb{P}\left(\xi(\theta^*, \alpha, H_T) = 1\right) \leq \alpha, \tag{27}$$

*i.e. type I error is asymptotically controlled.*

Condition (i) of Lemma 7 states that the quantiles of the simulated distribution for all sample paths $\omega$ are more extreme than that of the observed test statistic distribution. Condition (ii) is a technical condition for an application of the continuous mapping theorem, and is satisfied in most general cases (e.g. the distribution of $\rho(H_T)$ is dominated by the Lebesgue measure). Below, we show how these two conditions provide valid type I error control.

*Proof of Lemma 7.* We first write out the event in which we reject the null $\theta^*$, then leverage the Glivenko-Cantelli theorem and the dominated convergence theorem to obtain the desired results. Denoting $\hat{F}_B(\cdot)(\omega)$ as the simulated CDF with $B$ simulations, we obtain the following inequality:

$$\limsup_{T \to \infty} \lim_{B \to \infty} \mathbb{P}\left(\xi(\theta^*, \alpha, H_T) = 1\right) \tag{28}$$

$$= \limsup_{T \to \infty} \lim_{B \to \infty} \mathbb{P}\left(\left\{\hat{F}_B\left(\rho(H_T)\right)(\omega) \geq 1 - \alpha/2\right\} \cup \left\{\hat{F}_B\left(\rho(H_T)\right)(\omega) \leq 1 - \alpha/2\right\}\right) \tag{29}$$

$$= \limsup_{T \to \infty} \int_{\omega \in \Omega} \mathbf{1}\left[\left[\int \mathbf{1}\left[\rho(H_T) \geq \rho(H_T^{(i)})\right] \in [0, \alpha/2] \cup [1 - \alpha/2, 1] \, d\hat{F}(H_T^{(i)})(\omega)\right] \, d\omega \right. \tag{30}$$

Line 30 follows from (i) the dominated convergence theorem (DCT), (ii) the continuous mapping theorem (CMT), and (iii) the Glivenko-Cantelli theorem as $B \to \infty$. Note that because indicator functions are bounded, we can move the limit with respect to $B$ within the outer probability integral using DCT, and by condition (ii) in Lemma 7, we can pass this limit within the indicator function using CMT. By the fact that each $\rho(H_T^{(i)})$ is identically and independently distributed, as $B \to \infty$, the Glivenko-Cantelli Theorem states that our empirical CDF function $\hat{F}_B(\cdot)(\omega)$ converges to $\hat{F}(\cdot)(\omega)$ for all $\omega \in \Omega$. Further expanding our line 30, we obtain

$$\limsup_{T \to \infty} \lim_{B \to \infty} \mathbb{P}\left(\xi(\theta^*, \alpha, H_T) = 1\right) \tag{31}$$

$$\leq \limsup_{T \to \infty} \int_{\omega \in \Omega} \left(\mathbf{1}\left[\left[\int \mathbf{1}\left[\rho(H_T) \geq \rho(H_T^{(i)})\right] \in [0, \alpha/2] \, d\hat{F}(H_T^{(i)})(\omega)\right]\right]\right) d\omega \tag{32}$$

$$+ \int_{\omega \in \Omega} \left(\mathbf{1}\left[\left[\int \mathbf{1}\left[\rho(H_T) \geq \rho(H_T^{(i)})\right] \in [1 - \alpha/2, 1] \, d\hat{F}(H_T^{(i)})(\omega)\right]\right]\right) d\omega \tag{33}$$

$$= \limsup_{T \to \infty} \int_{\omega \in \Omega} \left(\mathbf{1}\left[\hat{F}(\rho(H_T))(\omega) \leq \alpha/2\right]\right) d\omega + \int_{\omega \in \Omega} \left(\mathbf{1}\left[\hat{F}(\rho(H_T))(\omega) \geq 1 - \frac{\alpha}{2}\right]\right) d\omega \tag{34}$$

$$= \limsup_{T \to \infty} \int_{\omega \in \Omega} \left(\mathbf{1}\left[\rho(H_T) \leq \hat{F}^{-1}\left(\alpha/2\right)(\omega)\right]\right) d\omega \tag{35}$$

$$+ \int_{\omega \in \Omega} \left(\mathbf{1}\left[\rho(H_T) \geq \hat{F}^{-1}\left(1 - \frac{\alpha}{2}\right)(\omega)\right]\right) d\omega \tag{36}$$

$$= \limsup_{T \to \infty} F\left(\sup_{\omega \in \Omega} \hat{F}^{-1}(\alpha/2)(\omega)\right) + \left(1 - F\left(\inf_{\omega \in \Omega} \hat{F}^{-1}(1 - \alpha/2)(\omega)\right)\right) \tag{37}$$

$$\leq \alpha. \tag{38}$$

$\square$

The inequality in line 32 follows directly from a union bound argument. Line 37 follows from the fact $F(\cdot)$ is a monotonically increasing function. The last line follows by condition (i) of Lemma 7, giving us the desired result.

### D.2.1 Proof for Example 1

We focus on two distinct cases: (i) $\mu_1^* \neq \mu_2^*$, and (ii) $\mu_1^* = \mu_2^*$. Let $\sigma_1^*, \sigma_2^*$ denote the true standard deviations of arm 1 and 2 respectively.

**Analysis of Case (i)** We first characterize the distribution of the observed sample mean test statistic under the ETC design for Case (i), keeping the true arm distributions $P_1, P_2$ fixed. The observed sample mean test statistic $\rho(H_T)$ converges to the following distribution:

$$\lim_{T \to \infty} \sqrt{T} \frac{(\rho(H_T) - \theta^*)}{\hat{\sigma}_1} \to_d \frac{2Z_1 + (\mathbf{1}\,[\mu_1^* \geq \mu_2^*])\,2\sqrt{2}Z_2}{1 + 2\,(\mathbf{1}[\mu_1^* \geq \mu_2^*])} \tag{39}$$

where $Z_i$ denotes a standard normal random variable. The standard normal random variables are a direct consequence of Slutksy's lemma using the the central limit theorem and $\hat{\sigma}_1 \to_P \sigma_1^*$ (shown in Lemma 6).The indicator term $\mathbf{1}[\mu_1^* \geq \mu_2^*]$ denotes the commit stage, where arm 1 is sampled $T/2$ additional times beyond the exploration stage. Because we assume $\mu_1 \neq \mu_2$, the indicator function $\mathbf{1}[\hat{\mu}_{T/2}(1) \geq \hat{\mu}_{T/2}(2)]$ converges to $\mathbf{1}[\mu_1^* \geq \mu_2^*]$ almost surely. This follows from the fact that $\mathbb{P}\left(\omega \in \Omega : \lim_{T \to \infty} \mathbf{1}[\hat{\mu}_{T/2}(1)(\omega) \geq \hat{\mu}_{T/2}(2)(\omega)] \neq \mathbf{1}[\mu_1^* \geq \mu_2^*]\right) = 0$ by the SLLN.

We now show that for every sample path $\omega \in \Omega$, the simulated distributions of $\rho(H_T^{(i)})$ uniformly protect type I error for $\theta_0 = \theta^*$. To show this, note that the distribution of the simulated test statistic $\rho(H_T^{(i)})(\omega)$, depends on the plug-in estimate $\hat{\mu}_2(\omega)$ used in the simulation.

We first note that the bias condition on $\epsilon_a$ in Theorem 1 states that $\sqrt{\frac{\log \log N_T(2)}{N_T(2)}}/\epsilon_a \to 0$. Under Assumption 1, $N_T(2) \to \infty$ almost surely (a.s.), so this condition permits two regimes: setting $\epsilon_a$ such that (a) bias term $\epsilon_a$ converges a.s. to zero, and (b) bias term $\epsilon_a$ does not converge a.s. to zero.

In setting (a), such as when $\epsilon_a = \frac{\log \log N_T(a)}{\sqrt{N_T(a)}}$, the test statistic $\rho(H_T^{(i)})(\omega)$ converges in distribution to RHS of Equation 39, matching the distribution of the true test statistic $\rho(H_T)$. This follows directly from the Slutsky's lemma, now including a vanishing term $\epsilon_a \to_{a.s.} 0$. As $B \to \infty$, the Glivenko-Cantelli theorem [29] ensures that the simulated distribution converges to that of the true distribution, ensuring type I error guarantees.

In setting (b), such as when $\epsilon_a$ is a positive constant, the simulated distribution $\rho(H_T^{(i)})$ for sample path $\omega \in \Omega$ takes the form in Equation 40. Note that our convergence in distribution denotes convergence of the simulated distribution (over the randomness generated by Algorithm 1) for a fixed sample path $\omega \in \Omega$ as $T \to \infty$.

$$\lim_{T \to \infty} \sqrt{T} \frac{\left(\rho(H_T^{(i)})(\omega) - \theta^*\right)}{\hat{\sigma}_1(\omega)} \to_d \frac{2Z_1 + (\mathbf{1}\,[\mu_1^* \geq \mu_2^* + \epsilon_2(\omega)])\,2\sqrt{2}Z_2}{1 + 2\,(\mathbf{1}\,[\mu_1^* \geq \mu_2^* + \epsilon_2(\omega)])}. \tag{40}$$

We now examine the limiting behavior of $\epsilon_2(\omega)$ as $T \to \infty$. If $\epsilon_2(\omega) < |\mu_1^* - \mu_2^*|$ as $T \to \infty$, then we obtain the same distribution as the RHS in Equation 39. If $\epsilon_2(\omega) \geq |\mu_1^* - \mu_2^*|$ as $T \to \infty$, then the distribution of $\rho(H_T^{(i)})$ still provides valid type I error guarantees.

To show this, consider the case where $\mu_1^* < \mu_2^*$. If $\epsilon_2(\omega) \geq \mu_2^* - \mu_1^*$, then the distribution of the observed test statistic $\rho(H_T)$ and sample-path dependent simulated distribution $\rho(H_T^{(i)})(\omega)$ match in distribution due to $\mathbf{1}[\mu_1^* \geq \mu_2^* + \epsilon_2(\omega)] = \mathbf{1}[\mu_1^* \geq \mu_2^*]$. In the case where $\mu_1^* > \mu_2^*$, if $\epsilon_2(\omega) \geq \mu_1^* - \mu_2^*$, the distribution of the observed test statistic obtains the limiting distribution

$$\lim_{T \to \infty} \sqrt{T} \frac{(\rho(H_T) - \theta^*)}{\hat{\sigma}_1} \to_d \frac{2Z_1 + 2\sqrt{2}Z_2}{3} =_d \frac{2Z_3}{\sqrt{3}}, \tag{41}$$

whereas the simulated test statistic distribution $\rho(H_T^{(i)})(\omega)$ for sample path $\omega \in \Omega$ takes the form

$$\lim_{T \to \infty} \sqrt{T} \frac{\left(\rho(H_T^{(i)})(\omega) - \theta^*\right)}{\hat{\sigma}_1(\omega)} \to_d 2Z_1 \quad \forall \, \omega \in \Omega. \tag{42}$$

The observed test statistic has a tighter limiting distribution around $\theta^*$ than our simulated test statistic, resulting in valid type I error control. By the Glivenko-Cantelli theorem [29] for uniform almost sure convergence for the empirical CDF (applied to the number of simulations $B$), we obtain the

following result, where $\mu$ is a dominating measure for the test statistic $\rho(H_T^{(i)})$.

$$\lim_{T \to \infty} \lim_{B \to \infty} \mathbb{P}\left(\left\{\hat{F}\left(\rho(H_T)\right) \geq 1 - \alpha/2\right\} \cup \left\{\hat{F}\left(\rho(H_T)\right) \leq 1 - \alpha/2\right\}\right) \tag{43}$$

$$= \lim_{T \to \infty} \mathbb{P}\left(\mathbf{1}\left[\left[\int \mathbf{1}\left[\rho(H_T) \geq \rho(H_T^{(i)})\right] \in [0, \alpha/2] \cup [1 - \alpha/2, 1]\right] d_\mu(\rho(H_T^{(i)}))\right) \tag{44}$$

$$\leq \lim_{T \to \infty} \mathbb{P}\left(\mathbf{1}\left[\left[\int \mathbf{1}\left[\rho(H_T) \geq \rho(H_T^{(i)})\right] \in [0, \alpha/2]\right] d_\mu(\rho(H_T^{(i)}))\right) \tag{45}$$

$$+ \lim_{T \to \infty} \mathbb{P}\left(\mathbf{1}\left[\left[\int \mathbf{1}\left[\rho(H_T) \geq \rho(H_T^{(i)})\right] \in [1 - \alpha/2, 1]\right] d_\mu(\rho(H_T^{(i)}))\right) \tag{46}$$

$$= \Phi\left(\sqrt{3}\Phi^{-1}(\alpha/2)\right) + \left(1 - \Phi\left(\sqrt{3}\Phi^{-1}(\alpha/2)\right)\right) \tag{47}$$

$$\leq \alpha/2 + \alpha/2 = \alpha, \tag{48}$$

where $\Phi(\cdot)$ and $\Phi^{-1}(\cdot)$ denote the CDF and quantile function of a standard random normal variable. Line 44 follows from the Glivenko-Cantelli theorem as $B \to \infty$ and the dominated convergence theorem applied to our indicator functions in order to move the limits within the outer probability integral. The inequality in line 46 follows from a simple union bound. Line 47 is a direct consequence of the limiting results in equations (41) and (42). Finally, line 48 follows from the fact that (i) $\Phi\left(\Phi^{-1}(\alpha/2)\right) = \alpha/2$, (ii) $\Phi(\cdot)$ is a monotone function, and (iii) $\sqrt{3}\Phi^{-1}(\alpha/2) \leq \Phi^{-1}(\alpha/2)$. This finishes the proof in the case where $\mu_1^* \neq \mu_2^*$.

**Analysis of Case (ii)** When $\mu_1^* = \mu_2^*$, the decision for the commit stage is nondeterministic as $T \to \infty$, resulting in the distribution of the observed test statistic $\rho(H_T)$ taking the following form:

$$\lim_{T \to \infty} \sqrt{T}\frac{(\rho(H_T) - \theta^*)}{\hat{\sigma}_1} \to_d \frac{2Z_1 + \left(\mathbf{1}\left[\sigma_1^* Z_1 \geq \sigma_2^* Z_3\right]\right) 2\sqrt{2}Z_2}{1 + 2\left(\mathbf{1}[\sigma_1^* Z_1 \geq \sigma_2^* Z_3]\right)}. \tag{49}$$

This result follows from similar arguments to the section above, but because $\mu_1^* = \mu_2^*$, we are left with only noise scaled by the standard deviations of each arm.

We now provide the limiting distribution for the simulated test statistic $\rho(H_T^{(i)})$ for all sample paths $\omega \in \Omega$. For a given sample path $\omega \in \Omega$, the limiting distribution takes the form

$$\lim_{T \to \infty} \sqrt{T}\frac{\left(\rho(H_T^{(i)})(\omega) - \theta^*\right)}{\hat{\sigma}_1(\omega)} \to_d 2Z_1 \quad \forall\, \omega \in \Omega. \tag{50}$$

We prove the result of Equation 50 below. This result is a direct consequence of our additional bias term $\epsilon_a$, which satisfies the condition $\left(\frac{\log \log N_T(a)}{N_T(a)}\right)/\epsilon_a \to 0$. At time $T/2$ in our simulation, we decide the arm 1 to sample an additional $T/2$ times based on the following indicator function (where 1 implies that we sample arm 1 $T/2$ additional times):

$$\mathbf{1}\left[\mu_1^*(\omega) + \frac{\hat{\sigma}_1(\omega)}{\sqrt{T/4}}Z_1 \geq \hat{\mu}_T(2)(\omega) + \epsilon_2(\omega) + \frac{\hat{\sigma}_2(\omega)}{\sqrt{T/4}}Z_2\right] \tag{51}$$

$$= \mathbf{1}\left[\sqrt{T}\left(\mu_1^* - \hat{\mu}_T(2)(\omega)\right) + 2\hat{\sigma}_1(\omega)Z_1 - 2\hat{\sigma}_2(\omega) \geq \sqrt{T}\epsilon_2(\omega)\right] \tag{52}$$

$$= \mathbf{1}\left[\underbrace{\sqrt{\frac{T}{N_T(2)(\omega)}}\frac{\sqrt{N_T(2)(\omega)}\left(\mu_1^* - \hat{\mu}_T(2)(\omega)\right)}{\sqrt{\log \log N_T(2)(\omega)}}}_{(a)} + \underbrace{\frac{2\hat{\sigma}_1(\omega)Z_1 - 2\hat{\sigma}_2(\omega)}{\sqrt{\log \log N_T(2)(\omega)}}}_{(b)} \geq \right. \tag{53}$$

$$\left.\underbrace{\sqrt{\frac{T}{N_T(2)(\omega)}}\frac{\epsilon_2(\omega)}{\sqrt{\frac{\log \log N_T(2)(\omega)}{N_T(2)(\omega)}}}}_{(c)}\right]. \tag{54}$$

We analyze the limits of terms (a), (b), and (c) below.

For term (a), we upper bound its limiting value and show that it must be finite. Note that $\frac{T}{N_T(2)(\omega)} \leq 2$ for all $\omega \in \Omega$, and $\lim_{T \to \infty} \frac{\sqrt{N_T(2)(\omega)}}{\sqrt{\log \log N_T(2)(\omega)}} |\mu_1^* - \hat{\mu}_T(2)(\omega)| \leq \sigma_2^* \sqrt{2}$ for all $\omega \in \Omega$ by Lemmas 3 and 4 and the assumption that $\mu_1^* = \mu_2^*$. Because we assume $\sigma_2^* < \infty$, term $(a)$ is upper bounded by the constant $\sigma_2^* \sqrt{2}$. For term (b), note that $\hat{\sigma}_1(\omega) \to \sigma_1^*$ and $\hat{\sigma}_2(\omega) \to \sigma_2^*$ for all $\omega \in \Omega$, meaning that the scalar values in the numerator are finite. Because $N_T(2)(\omega) \to \infty$ for all $\omega \in \Omega$, term $(b) \to 0$ for all $\omega \in \Omega$. Lastly, for term (c), we use the condition that $\left( \epsilon_2(\omega) / \left( \sqrt{\frac{\log \log N_T(2)(\omega)}{N_T(2)(\omega)}} \right) \right)^{-1} \to 0$ for all $\omega \in \Omega$, which implies that $(c)$ diverges to infinity. As a result, $(c) \to \infty$ almost surely.

Putting these pieces together, we obtain that our indicator function is equal to 0 as $T \to \infty$ for all sample paths $\omega \in \Omega$. As a result, we obtain as $T \to \infty$, for all realized sample paths $\omega \in \Omega$, our simulated sample paths select arm 2 as the arm to sample an additional $T/2$ times, resulting in the distribution provided in Equation 50.

We now show that the CDF of our simulated test statistic provides valid type I error control using Lemma 7. We start with condition (i). Let $\phi = \sqrt{T} \left( \rho(H_T) - \theta^* \right) / 2\hat{\sigma}_1$ and $\phi(\omega) = \sqrt{T} \left( \rho(H_T^{(i)})(\omega) - \theta^* \right) / 2\hat{\sigma}_1(\omega)$. Let $c = \Phi^{-1}(\alpha/2)$, $c' = \Phi^{-1}(1 - \alpha/2)$. Let $F(\cdot)$ and $\hat{F}^{-1}(\cdot)(\omega)$ denote the limiting CDF and inverse CDF of $\phi$ and $\phi(\omega)$ as $T \to \infty$ respectively. We show that $F(c) + 1 - F(c') \leq \alpha$. We start with $F(c)$, deriving an expression equivalent to $F(c) - \alpha/2$.

$$F(c) = \mathbb{P}(\phi \leq c) \tag{55}$$

$$= \frac{1}{2} \mathbb{P} \left( Z_1 \leq c \middle| Z_1 < \frac{\sigma_2^*}{\sigma_1^*} Z_3 \right) + \frac{1}{2} \mathbb{P} \left( \frac{Z_1 + \sqrt{2} Z_2}{3} \leq c \middle| Z_1 \geq \frac{\sigma_2^*}{\sigma_1^*} Z_3 \right) \tag{56}$$

Note that $\mathbb{P}(Z_1 \leq c) = \alpha/2$ by definition. We subtract $\mathbb{P}(Z_1 \leq c) = \alpha/2$ to $F(c)$ below:

$$F(c) - \alpha/2 = F(c) - \frac{1}{2} \left( \mathbb{P} \left( Z_1 \leq c \middle| Z_1 < \frac{\sigma_2^*}{\sigma_1^*} Z_3 \right) + \mathbb{P} \left( Z_1 \leq c \middle| Z_1 \geq \frac{\sigma_2^*}{\sigma_1^*} Z_3 \right) \right) \tag{57}$$

$$= \frac{1}{2} \left( \mathbb{P} \left( \frac{Z_1 + \sqrt{2} Z_2}{3} \leq c \middle| Z_1 \geq \frac{\sigma_2^*}{\sigma_1^*} Z_3 \right) - \mathbb{P} \left( Z_1 \leq c \middle| Z_1 \geq \frac{\sigma_2^*}{\sigma_1^*} Z_3 \right) \right) \tag{58}$$

$$= \frac{1}{2} \mathbb{P} \left( \frac{Z_1 + \sqrt{2} Z_2}{3} \leq c \leq Z_1 \middle| Z_1 \geq \frac{\sigma_2^*}{\sigma_1^*} Z_3 \right) \tag{59}$$

$$- \frac{1}{2} \mathbb{P} \left( \frac{Z_1 + \sqrt{2} Z_2}{3} \geq c \geq Z_1 \middle| Z_1 \geq \frac{\sigma_2^*}{\sigma_1^*} Z_3 \right) \tag{60}$$

$$= \frac{1}{2} \int_x \mathbb{P} \left( \frac{\sqrt{2} Z_2}{3} \leq c \leq \frac{2 Z_1}{3} \middle| Z_1 \geq \frac{\sigma_2^*}{\sigma_1^*} Z_3, \, Z_3 = x \right) df(x) \tag{61}$$

$$- \frac{1}{2} \int_x \mathbb{P} \left( \frac{\sqrt{2} Z_2}{3} \geq c \geq \frac{2 Z_1}{3} \middle| Z_1 \geq \frac{\sigma_2^*}{\sigma_1^*} Z_3, \, Z_3 = x \right) df(x) \tag{62}$$

$$\tag{63}$$

Note that line 58 follows from the expansion of $F(c)$ provided in line 56, and line 62 adds an additional conditioning statement to fix the value of $Z_3$, a standard normal random variable, with an additional integral over the standard normal distribution density $df(x)$. Leveraging the fact that (i) the truncated standard normal distribution with bounds $[a, b]$ has CDF $(\Phi(x) - \Phi(a)) / (\Phi(b) - \Phi(a))$ evaluated at $x$ and (ii) $Z_i$'s are all independent, we obtain

$$F(c) - \alpha/2 = \frac{1}{2} \int_x \mathbb{P}\left( \frac{\sqrt{2}Z_2}{3} \leq c \leq \frac{2Z_1}{3} \;\middle|\; Z_1 \geq \frac{\sigma_2^*}{\sigma_1^*} Z_3, \; Z_3 = x \right) df(x) \tag{64}$$

$$- \frac{1}{2} \int_x \mathbb{P}\left( \frac{\sqrt{2}Z_2}{3} \geq c \geq \frac{2Z_1}{3} \;\middle|\; Z_1 \geq \frac{\sigma_2^*}{\sigma_1^*} Z_3, \; Z_3 = x \right) df(x) \tag{65}$$

$$= \frac{1}{2} \int_x \Phi\left( \frac{3c}{\sqrt{2}} \right) \left( 1 - \frac{\Phi(3c/2) - \Phi(\sigma_2^* x/\sigma_1^*)}{1 - \Phi(\sigma_2^* x/\sigma_1^*)} \right) \tag{66}$$

$$- \left( 1 - \Phi\left( \frac{3c}{\sqrt{2}} \right) \right) \left( \frac{\Phi(3c/2) - \Phi(\sigma_2^* x/\sigma_1^*)}{1 - \Phi(\sigma_2 x/\sigma_1)} \right) df(x) \tag{67}$$

$$= \frac{1}{2} \int_x \left( \frac{\Phi(3c/2)}{1 - \Phi(\sigma_2^* x/\sigma_1^*)} \right) \left( 1 - 2\Phi\left( \frac{3c}{2} \right) + \Phi\left( \frac{\sigma_2^* x}{\sigma_1^*} \right) \right) df(x) \tag{68}$$

Repeating the same steps, we obtain that $(1 - F(c')) - \alpha/2$ is equivalent to

$$(1 - F(c')) - \alpha/2 = \int_x \frac{1}{2} \frac{\Phi(3c'/2)}{1 - \Phi(\sigma_2^* x/\sigma_1^*)} \left( 2\Phi\left( \frac{3c'}{2} \right) - \Phi\left( \frac{\sigma_2^* x}{\sigma_1^*} \right) - 1 \right) df(x). \tag{69}$$

Combining these two expressions, we obtain $F(c) + (1 - F(c')) - \alpha$ is equivalent to

$$\frac{1}{2} \int_x \frac{\Phi(3c/2)\Phi(3c'/2)}{1 - \Phi(\sigma_2^* x/\sigma_1^*)} \left( \frac{(\Phi(3c/2) - \Phi(3c'/2))(1 + \Phi(\sigma_2^* x/\sigma_1^*)) + 2\left(\Phi(3c'/2)^2 - \Phi(3c/2)^2\right)}{\Phi(3c/2)\Phi(3c'/2)} \right) df(x) \tag{70}$$

Consider the numerator of the second term within the integral. Note that $\Phi(3c/2) \leq \Phi(3c'/2)$ by definition of $c = \Phi^{-1}(\alpha/2)$ and $c' = \Phi^{-1}(1 - \alpha/2)$. From this simple observation, we obtain that the first and second terms in the numerator are nonpositive and nonnegative, i.e.

$$(\Phi(3c/2) - \Phi(3c'/2)) \leq 0, \tag{71}$$

$$(1 + \Phi(\sigma_2^* x/\sigma_1^*)) + 2\left(\Phi(3c'/2)^2 - \Phi(3c/2)^2\right) \geq 0, \tag{72}$$

which provides the desired result that

$$F(c) + (1 - F(c')) - \alpha = \tag{73}$$

$$\frac{1}{2} \int_x \frac{\Phi(3c/2)\Phi(3c'/2)}{1 - \Phi(\sigma_2^* x/\sigma_1^*)} \tag{74}$$

$$\left( \frac{\underbrace{(\Phi(3c/2) - \Phi(3c'/2))}_{\geq 0} \underbrace{(1 + \Phi(\sigma_2^* x/\sigma_1^*)) + 2\left(\Phi(3c'/2)^2 - \Phi(3c/2)^2\right)}_{\leq 0}}{\Phi(3c/2)\Phi(3c'/2)} \right) df(x) \tag{75}$$

$$\leq 0. \tag{76}$$

Now, note that by Equation 50, we know that the following holds:

$$\lim_{T \to \infty} \sup_{\omega \in \Omega} \hat{F}^{-1}(\alpha/2)(\omega) = \Phi^{-1}(\alpha/2), \quad \lim_{T \to \infty} \inf_{\omega \in \Omega} \hat{F}^{-1}(1 - \alpha/2)(\omega) = \Phi^{-1}(1 - \alpha/2). \tag{77}$$

Thus, as $T \to \infty$, by the dominated convergence theorem (which applies for any $\alpha \in (0, 1)$),

$$\limsup_{T \to \infty} F\left( \sup_{\omega \in \Omega} \hat{F}^{-1}(\alpha/2)(\omega) \right) + \left( 1 - F\left( \inf_{\omega \in \Omega} \hat{F}^{-1}(1 - \alpha/2)(\omega) \right) \right) \tag{78}$$

$$= F(c) + (1 - F(c')) \leq \alpha. \tag{79}$$

This verifies condition (i). Condition (ii) is satisfied by the fact that the probability of $\rho(H_T)$ taking two specific values has probability 0 due to $\rho(H_T)$ being a random variable dominated by the Lebesgue measure. Therefore, by Lemma 7, we have type I error control.

**Proof for Example 2** We prove this result by first leveraging stability results for UCB presented by Khamaru and Zhang [18]. For completeness, we provide this result below in Lemma 8

**Lemma 8** (Theorem 3.1 of Khamaru and Zhang [18])**.** *Denote* $\Delta_a = (\max_{a \in [K]} \mu_a^* - \mu_a^*)$, *and assume that the following conditions hold:*

*(i)* $0 \leq \Delta_a/\sqrt{2 \log T} = o(1)$ *for all* $a \in [K]$,

*(ii)* $P_a$ *is* $\lambda_a$*-subgaussian for all arms* $a \in [K]$, *where* $|\lambda_a| \leq B$ *for some constant* $B$.

*Then,* $\frac{N_T(a)}{(1/\sqrt{N^*}+\sqrt{\Delta_a^2/2 \log T})^{-2}} \to_p 1$, *where* $N^*$ *is defined as the unique solution to the following:*

$$\sum_{a \in [K]} \frac{1}{\left( \sqrt{T/N^*} + \sqrt{T\Delta_a^2/2 \log T} \right)^2} = 1. \tag{80}$$

We show that under our simulation procedure, for all possible experiment sample paths $\omega \in \Omega$, the number of samples $N_T(1)(\omega)$ converges in probability to a smaller limiting value than under the true vector of means $\boldsymbol{\mu}^*$, resulting in larger upper (and smaller lower) quantiles using Lemma 5.

By Corollary 1, there exists a $T(\omega)$ for all $\omega \in \Omega$ such that $\hat{\mu}_a(\omega) \geq \mu_a^*$ for all $a \neq 1$. Let $\hat{\Delta}_a(\omega) = \max\{\theta^*, \max_{a' \in [K]\backslash\{1\}} \hat{\mu}_T(a')(\omega) + \epsilon_a(\omega)\} - \mathbf{1}[a \neq 1] (\hat{\mu}_T(a') + \epsilon_a(\omega)) - \mathbf{1}[a = 1]\theta^*$, and let $\Delta_a$ denote the ground truth equivalent with respect to true mean vector $\boldsymbol{\mu}^*$. By Lemma 8, we obtain that the number of arm pulls $N_T(1)(\omega)$ under simulations of $\rho(H_T^{(i)})$ satisfies the following limit:

$$\frac{N_T(1)(\omega)}{\left(1/\sqrt{N^*(\omega)} + \sqrt{\hat{\Delta}_a^2(\omega)/2 \log T}\right)^{-2}} \to_p 1, \quad \sum_{a \in [K]} \frac{1}{\left( \sqrt{T/N^*(\omega)} + \sqrt{T\hat{\Delta}_a^2/2 \log T} \right)^2} = 1. \tag{81}$$

Because $\liminf_{T \to \infty} \hat{\mu}_a(\omega) \geq \mu_a^*$ for all $a \neq 1$ and $\omega \in \Omega$, $\limsup_{T \to \infty} \frac{\left(1/\sqrt{N^*(\omega)}+\sqrt{\hat{\Delta}_a^2(\omega)/2 \log T}\right)^{-2}}{\left(1/\sqrt{N^*}+\sqrt{\Delta_a^2/2 \log T}\right)^{-2}} \leq 1$ for all $\omega \in \Omega$, where $N^*$ is as defined in Lemma 8. As a result, for each $\omega \in \Omega$, (i) $\frac{N_T(1)(\omega)}{\left(1/\sqrt{N^*(\omega)}+\sqrt{\hat{\Delta}_a^2(\omega)/2 \log T}\right)^{-2}} \to_p 1$, and (ii) $\limsup_{\omega \in \Omega, T \to \infty} N_T(1)(\omega) / \left(1/\sqrt{N^*} + \sqrt{\Delta_a^2/2 \log T}\right)^{-2} \leq 1$. This implies that for each simulation under $\omega$, the distribution of $\rho(H_T^{(i)})(\omega)$ is asymptotically normal and centered at $\mu_1^*$, but has larger limiting variance than $\rho(H_T^{(i)})$. By the same argument (i.e. for all $\omega \in \Omega$, larger $1 - \alpha/2$ quantiles, smaller $\alpha/2$ quantiles for $\rho(H_T^{(i)})(\omega)$) as case (ii) in Example 1, we preserve type I error.

**Proof for Example 3** Example 3 proceeds similarly Example 2. First, we show that under this design, the test statistic $\rho(H_T)$ is asymptotically normal. Second, we show that simulated distribution of $\rho(H_T^{(i)})(\omega)$ has wider quantiles as a result of (i) lower limiting sampling rates and (ii) an asymptotically normal distribution for all $\omega \in \Omega$.

First, let $a^* = \operatorname{argmax}_{a \in [K]} \mu_a^*$, where $a^*$ is assumed to be unique by definition. For all arms $a \neq a^*$, note that $\frac{N_T(a)}{T\gamma/K} \to_p 1$, and for $a = a^*$, $\frac{N_T(a)}{T(1-\gamma)+T\gamma/K} \to_p 1$. This follows from the fact that we assume suboptimal arms are selected at a smaller rate than $c/T$ for some constant $c$ at each timestep, resulting in

$$\limsup_{T \to \infty} \frac{N_T(a)}{T} \leq \lim_{T \to \infty} \frac{\gamma T/K + c\log(T)}{T} \to_p \gamma/K \quad \forall\, a \neq a^*. \tag{82}$$

Likewise, an equivalent lower bound for the leftmost expression above is obtained by ignoring all samples of arm $a$ that are not obtained through forced exploration with probability $\gamma$. We obtain the expression for $a = a^*$ by noting that $\sum_{a \in [K]} \frac{N_T(a)}{T} = 1$ by definition. Given that the number of arm pulls converge to a constant (dependent on sample path) in probability, Lemma 5 states that $\sqrt{N_T(1)}\rho(H_T)/\hat{\sigma_1} \to_d N(0, 1)$. With positive bias term $\epsilon_a$ added to all other means, if $\epsilon_a \to 0$ as

$T \to \infty$, then the best arm does not change, and therefore the limiting distribution for $\rho(H_T^{(i)})$ and the limiting distribution for observed test statistic $\rho(H_T)$ are identical, and therefore we preserve type I error. If $\lim_{T \to \infty} \epsilon_a > 0$, then note that the only change in distribution may be that arm 1 goes from the best arm under $\boldsymbol{\mu}^*$ to a suboptimal arm under estimated nuisances $\hat{\boldsymbol{\mu}} = [\theta^*, \hat{\mu}_2, ..., \hat{\mu}_K]$. If this change does occur, then the simulated distribution $\rho(H_T^{(i)})$ has larger $1 - \alpha/2$ quantiles and smaller $\alpha/2$ quantiles as $T, B \to \infty$ due to (i) a centered normal distribution where (ii) standard deviations scale with the number of arm pulls. By the same argument as case (ii) in Example 1, we preserve type I error.

### D.3  Proof of Lemma 2

Before showing the proof of this approach, we first note a minor clarification that will be added to the main body in Remark 7. We assume the change in Remark 7 in our proof.

**Remark 7.** *Lemma 2 assumes that $\hat{C}(\alpha)$ is not the empty set almost surely - we will make this explicit in the next possible edit by modifying $\hat{C}(\alpha)$ to always include the empirical mean estimate $\hat{\mu}_T(1)$.*

The proof of Lemma 2 follows directly from Lemma 1. We prove this result using proof by contradiction, showing that the upper and lower bounds of our confidence set must converge to $\theta^*$ almost surely. To begin, we start with the lower bound $\hat{L}(\alpha) = \min\{\theta : \theta \in \hat{C}(\alpha)\}$.

To contradict $\hat{L}(\alpha) \to_{a.s.} \theta^*$, we assume that there exists a sample path $\omega \in \Omega$ (where $\mathbb{P}(\Omega) = 1$) and $\epsilon > 0$, such that $\lim_{T \to \infty} |\hat{L}(\alpha)(\omega) - \theta^*| > \epsilon$. However, by Lemma 1, for all $\omega \in \Omega$, there exists a $T(\omega)$ such that for $\hat{L}(\alpha)(\omega) \neq \theta^*$, $\hat{F}(\rho(H_T)(\omega)) \in \{0, 1\}$ for all $T > T(\omega)$, meaning that $\hat{L}(\alpha)(\omega) \notin \hat{C}(\alpha)$ by definition in Algorithm 2. This results in a contradiction, demonstrating that $\hat{L}(\alpha) \to_{a.s.} \theta^*$. We repeat this proof for $\hat{U}(\alpha) = \max\{\theta : \theta \in \hat{C}(\alpha)\}$ to obtain analogous results, proving that $\hat{C}(\alpha) \to_{a.s.} \{\theta^*\}$. Because $\hat{\theta} \in \hat{C}(\alpha)$ and $\hat{C}(\alpha) \to_{a.s.} \{\theta^*\}$, we obtain $\hat{\theta} \to_{a.s.} \theta^*$.

## E  Limitations and Broader Impacts

**Limitations**  The key limitation of this approach is that (i) there exists minimal unifying theory on what/which designs this approach provides valid type I error and (ii) its relatively high computational expense compared with standard inference approaches. As shown in Appendix D, for each example in Theorem 1, we verify the examples on a case-by-case basis. It would be of great interest to have a unifying condition on designs and arm instances under which this approach provides valid inference. Furthermore, our approach, while maintaining a reasonable level of computational tractability relative to the grid-scanning approach discussed in Section 3, is far more expensive than methods such as a Wald confidence interval. Thus, our approach is best suited for settings where experiment horizons are moderate and one wishes to conduct inference on treatments/arms not specifically targeted by the design. Our approach provides the largest gains for target parameters not targeted by the design (e.g. arms with relatively low means), and should be used in settings where inference on all options offered throughout the experiment is desired/necessary.

**Broader Impacts**  Our work contributes to the literature on inference post adaptive experimentation, and is among the few works (to the best of our knowledge) that aims to use a computational approach to conduct inference in this setting. Much like how bootstrap approaches generally tend to outperform asymptotic or finite-sample inference based on Gaussian approximations or concentration inequalities, our empirical results suggest our simulation-based approach may improve power and reduce confidence interval widths for many classical settings. Furthermore, our work relaxes conditions such as conditional positivity, while maintaining asymptotic control of type I error. We note that our guarantees on error rates is asymptotic, and therefore caution practitioners who hope to use our approach when sample sizes are overly small.

