# OpenReview forum: "Simulation-Based Inference for Adaptive Experiments"
_NeurIPS.cc/2025/Conference — NeurIPS 2025 poster_

### Official Review · Reviewer_Us9g · 2025-06-25

**Clarity:** 3
**Significance:** 2
**Originality:** 3
**Rating:** 4
**Confidence:** 3

**Summary:**

The paper introduces a novel method for conducting statistical inference—hypothesis testing and confidence intervals—after adaptive experimental designs such as multi-arm bandits. Specifically, it proposes an optimistically biased nuisance estimates, a technique called simulation with optimism, which helps preserve type I error control even when standard plug-in estimators fail. As a result, it avoids some standard positivity assumptions, applicable to a broad range of adaptive designs, such as explore-then commit (ETC) and upper-confidence-bound style (UCB) algorithms. It offers asymptotic guarantees for type I error, confidence coverage, and point estimate consistency, and achieves up to 50% narrower confidence intervals, particularly for under-sampled arms, compared to existing approaches, as demonstrated on both synthetic and real-world data.

**Questions:**

Please see 3 bullet points in Section Weakness.

**Ethical Concerns:**

["NO or VERY MINOR ethics concerns only"]

**Final Justification:**

I would recommend a score of 4 (borderline accept) to this paper as the authors' response solved my questions in the review and provided very detailed literature comparison of other existing methods.

**Limitations:**

yes

**Quality:**

3

**Strengths And Weaknesses:**

Strength: The method avoids some standard positivity assumptions, applicable to a broad range of adaptive designs, such as explore-then commit (ETC) and upper-confidence-bound style (UCB) algorithms, with theoretical guarantees. Simulations of this method showed a very big improvements on the width of confidence intervals up to 50%.

Weakness:
1. The algorithm assumes the experiment policy is known to simulate actions while reweighting or martingale-based methods can work solely with the observed data, without simulating from the policy.

2. For each null hypothesis, the method re-simulates the full trajectory of the experiment.

3. The confidence intervals are constructed by inverting pointwise hypothesis tests over a grid of possible values without a closed form unlike some other method such as reweighting-based asymptotic inference.

Thus, in general, while the paper proposed a new method that avoids problems in the existing literatures, it also causes some other main problems.

---

> ### Author Rebuttal · Authors · 2025-07-28
>
> We thank the reviewer for their detailed reading of our work. In particular, we appreciate that the reviewer found our method applicable to a broad range of experiment designs and empirically well-performing with significantly tighter confidence intervals. We hope to address the questions, comments, and clarifications raised by the reviewer below.
>
> ***Knowledge of the Experiment Policy***: We note that knowledge of the experiment policy is a *standard assumption for after-study inference in adaptive experiments.* Even when using reweighting/martingale methods, knowledge of the policy (i.e. full knowledge of conditional propensity scores at each time $t \in \mathbb{N}$) is required to implement existing inference approaches. Zhang et. al. (2021) use a reweighted estimator based on the martingale central limit theorem (MCLT) that directly leverages the true propensity scores $\pi_t$ in each term of the summation, and propose a weighting scheme that also incorporates the true propensity scores. Similarly, Bibaut et. al. (2021) use a different reweighting approach for a similar inference approach based on the MCLT and still require knowledge of the true propensity scores at each time $t \in \mathbb{N}$ to construct their confidence interval. Lastly, the approach of Hadad et. al. (2021) builds upon unbiased scoring rules that leverage the propensity scores in the denominator (similar to the augmented inverse propensity weighted (AIPW) approach), and assume that the conditional propensities (and therefore the sampling scheme) are known. In conclusion, approaches that leverage the MCLT through reweighing do indeed require knowledge of the underlying data collection scheme $\pi$, just like our approach here.
>
> **To make this clear, we will provide this discussion as a remark in Section 3. We hope that this clarifies that while our approach uses knowledge of the sampling scheme used for data collection, other approaches also require this assumption.**
>
> ***Costs of Simulation***: As the reviewer notes, a key difference between our approach and existing works is the need to simulate the experiment in order to construct our confidence intervals and hypothesis tests. While this does result in additional computational expenses, we (i) provide computational results regarding the cost of our procedure in Appendix C.1 of our work and (ii) note that other inference approaches often require resimulation of the original data collection policy for implementation.
>
> *Runtimes*: Our runtime results in Appendix C.1 show that to construct our confidence intervals run in reasonable times for moderate values of $G$, the size of the grid, and $B$, the number of simulations per null value. Even when run locally, our confidence intervals take less than a minute to construct. In applications such as political science (shown in our case study example) or healthcare, where data collection is expensive/time-consuming and tight inference is of paramount importance, we expect that runtime of our approach will not be the bottleneck for data analysis. We mention that Reviewer PfWo notes this issue is likely to be less of a concern in such settings.
>
> *Other approaches actually resimulate as well.* As discussed above, reweighing/MCLT approaches for inference on adaptively collected data require knowledge of the true conditional propensity scores at each time $t \in \mathbb{N}$. To obtain the true conditional propensity scores under popular sampling schemes such as Thompson sampling (which does not have clear, closed-form propensity scores such as $\epsilon$-greedy sampling schemes), works such as [1] and [3] resimulate the sampling policy $\pi$ using history $H_{t-1}$ to obtain conditional propensity scores $\pi_t$ at each time point. We emphasize that while the reweighing/MCLT approaches do not directly require resimulation, it is common practice to estimate the true propensity scores needed for reweighing approaches by simulation the known sampling policy, as done in works such as [1] and [3].
>
> ***Limitations of Grids / Lack of Closed Form Solutions***: We acknowledge that our approach does not have a closed-form solution and relies on a grid-search approach to form an approximate confidence interval. However, a lack of closed-form solution and the use of grids need not limit our work. Recent works for constructing confidence intervals and confidence sequences rely on a similar approach. For example, the GRAPA/ONS/Hedged betting-style confidence sequences by Waudby-Smith et. al. (2021) rely on a similar grid approach for testing null values of the mean in the anytime-valid inference setting. Despite lack of a closed form solution, these approaches demonstrate significant improvements to confidence sequence widths, much like the gains observed by our method for fixed-time inference on adaptively collected data. We emphasize that while closed-form solutions are indeed desirable for any method, approaches without closed-form solutions, such as our proposed method, may provide more powerful inference for the data at hand.
>
> **To ensure this point is well-communicated, we will add this remark in Section 3. We thank the reviewer for this clarifying point.**
>
> ***Difference in Applications with Reweighing/Martingale Methods***: Our approaches are not intended to replace the reweighing/MCLT inference approaches for adaptively collected data. Rather, they enable valid inference in settings where *inference is not feasible using these existing methods*. For example, examples 1 and 2 in our work do not permit inference using reweighing/MCLT approaches due to having conditional propensity scores of 0 (and thus suffer from terms diverging to infinity in their estimator). Because such approaches *cannot* provide valid inference in such settings, analysts are often left with only the option of anytime valid inference (which is conservative as shown in our empirical results). Our work enables fixed-time testing with similar guarantees as MCLT methods under experiment designs that *do not support reweighing/MCLT approaches*. As such, our approach is not exactly comparable with MCLT methods due to the latter's inability to provide valid inference in the designs presented in our work. While our method presents additional computational expense, it enables fixed-time inference in settings where MCLT/reweighing estimators are guaranteed to fail and will not work.
>
> **To make this point clear, we will add additional details in our related work (Section 1.1) to stress these points.**
>
> We thank the reviewer for raising these points - they have added valuable clarity to our work. We will incorporate these clarifications (as mentioned above) into our final draft, and hope that this addresses any questions or comments the reviewer may have.  **We hope our responses have addressed your key concerns and would be grateful if you would consider a higher score in light of these clarifications.** Please feel free to ask further questions if any remain!
>
> [1] K. W. Zhang, L. Janson, and S. A. Murphy. Statistical inference with m-estimators on adaptively collected data, 2021.
>
> [2] A. Bibaut, A. Chambaz, M. Dimakopoulou, N. Kallus, and M. van der Laan. Post-contextual- bandit inference, 2021.
>
> [3] V. Hadad, D. A. Hirshberg, R. Zhan, S. Wager, and S. Athey. Confidence intervals for policy evaluation in adaptive experiments. Proceedings of the National Academy of Sciences, 118(15): e2014602118, 2021
>
> [4]I. Waudby-Smith and A. Ramdas. Estimating means of bounded random variables by betting, 2021.

---

> > ### Comment · Reviewer_Us9g · 2025-08-04
> >
> > I thank the authors for providing such a detailed response to my questions. My main concern about this paper (as it is more of a methodology one) is on the comparison with other literatures as this topic is in an active area and there are many other existing papers/methods, where each of them flavors their own approach. Thus, I am very happy to see and learn from this detailed clarifications on the difference/relevance of the authors' approach with other approaches. I do appreciate the fact that empirically this method demonstrated a very good improvement on the confidence interval. Thus, I would raise my score given the authors could insert a remark/subsection into the paper about this comparison to other methods with more details.

---

> > > ### Author Response · Authors · 2025-08-05
> > >
> > > We are glad that the reviewer found our clarifications detailed regarding comparisons with other alternative approaches.
> > >
> > > To be specific, we *will* include the following remarks and clarifications in the final draft (regarding comparisons/aims relative to existing approaches):
> > > * Remark 1 (at the end of Section 2, before Section 2.1): We note that knowledge of the experiment policy is a standard assumption for after-study inference in adaptive experiments. Even when using reweighting/martingale methods, knowledge of the policy (i.e. full knowledge of conditional propensity scores at each time ) is required to implement existing inference approaches. Zhang et. al. (2021) use a reweighted estimator based on the martingale central limit theorem (MCLT) that directly leverages the true propensity scores in each term of the summation, and propose a weighting scheme that also incorporates the true propensity scores. Similarly, Bibaut et. al. (2021) and Hadad et. al. (2021) use a different reweighing approach for a similar inference approach based on the MCLT and still require knowledge of the true propensity scores at each time  to construct their confidence interval. In conclusion, approaches that leverage the MCLT through reweighing require knowledge of the underlying data collection scheme , just like our approach here.
> > >
> > > * Added Paragraph in Conclusion: Our approaches are not intended to replace the reweighing/MCLT inference approaches for adaptively collected data. Rather, they enable valid inference in settings where inference is not feasible using these existing methods. For example, examples 1 and 2 in our work do not permit inference using reweighing/MCLT approaches due to having conditional propensity scores of 0 (and thus suffer from terms diverging to infinity in their estimator). Because such approaches cannot provide valid inference in such settings, analysts are often left with only the option of anytime valid inference (which is conservative as shown in our empirical results). Our work enables fixed-time testing with similar guarantees as MCLT methods under experiment designs that do not support reweighing/MCLT approaches. As such, our approach is not exactly comparable with MCLT methods due to the latter's inability to provide valid inference in the designs presented in our work. While our method presents additional computational expense, it enables fixed-time inference in settings where MCLT/reweighing estimators are guaranteed to fail and will not work.
> > >
> > > These additions, as well as all other clarifying remarks in the rebuttal (regarding computational costs of grid-based testing, runtime, closed-form solutions) will be incorporated into our final manuscript.
> > >
> > > We thank the reviewer for their insightful comments. We hope our responses have addressed your key concerns (inclusion of discussions regarding other approaches) and would be grateful if you would consider a higher score in light of these clarifications/additions!

---

### Official Review · Reviewer_4paL · 2025-06-28

**Clarity:** 3
**Significance:** 3
**Originality:** 3
**Rating:** 4
**Confidence:** 4

**Summary:**

This paper studies multi-armed bandit experimental designs that relax the common requirement for sampling schemes to assign nonzero probability to each option at every time step. The authors propose a simulation-based method for hypothesis testing and for constructing confidence intervals for parameters of interest. The approach leverages additional trajectories generated based on the optimistic principle to facilitate the construction of $p$-values and confidence intervals. Theoretical guarantees for the proposed method are provided.

**Questions:**

- The current setting only considers $(A_i, X_i)$. However, in many practical applications, experimental designs often depend on additional contextual information, such as $Z_i$. It would be valuable to discuss whether and how the proposed method can be extended to accommodate settings involving such additional contextual variables.

- Regarding the historical data $H_T$, the proposed method relies on knowledge of the sampling scheme. In practice, the exact sampling scheme may be unknown. How the method can be applied in such cases, either from a theoretical perspective or through empirical evaluation.

**Ethical Concerns:**

["NO or VERY MINOR ethics concerns only"]

**Final Justification:**

This paper develops a simulation-based method for hypothesis testing and confidence interval construction in multi-armed bandit designs that need not sample all options at each step. The approach uses optimistic trajectories to compute $p$-values and intervals, with theoretical guarantees and asymptotic strong consistency, and applies to a broad class of experimental designs.  While I appreciate the authors’ clarifications and willingness to revise their work during the rebuttal, I am maintaining my original score of Weak Accept, as I am still unable to fully determine whether the paper meets the accept threshold.

**Limitations:**

See the Weaknesses and Questions sections for limitations.

**Paper Formatting Concerns:**

- It would be helpful to provide more justification or explanation for the statement in line 34 that “standard hypothesis tests and confidence intervals are not guaranteed to provide their nominal error control.”

- In Definition 2, it would be more precise to state that the (random) confidence interval covers the true parameter $\theta^*$.

- In Algorithm 1, it it $\widehat{\mu}_1 = \theta_0$?

- It would be helpful to provide more detailed derivations for Equation (27)  in the Appendix

**Quality:**

3

**Strengths And Weaknesses:**

Strengths
- The proposed test is applicable to a broad class of experimental designs, and the authors establish the asymptotic strong consistency of the proposed estimator.


Weaknesses
- In Lemma 1, it appears somewhat counterintuitive that the proposed test can detect alternatives at any separation rate. For example, in the standard $t$-test under a normal distribution, the power converges to 1 only when the difference between the true mean and the null mean exceeds the typical $T^{-1/2}$ separation rate.

- It would be helpful to clarify whether the result in Lemma 1 also applies to the test incorporating optimism, as presented in Theorem 1.

- It is unclear whether the theoretical guarantees in Theorem 1 extend to other experimental designs, such as simple randomized designs.

- Is the proposed method any time valid?

- For the case of "Arm 2 CA, 2 BM," the proposed confidence intervals appear wider than those obtained from the competing methods.

---

> ### Author Rebuttal · Authors · 2025-07-29
>
> We thank the reviewers for their detailed comments and questions, particularly regarding the theoretical results of our work. We hope to address any confusions and clarify our results below.
>
> ***Questions Regarding Lemma 1***:
> We apologize for any confusion regarding Lemma 1. In particular, we mean that for a *fixed* null $\theta_0 \neq \theta^{\*}$, our test rejects this null with probability 1 as $T \rightarrow \infty$. In particular, this result applies for a fixed null value $\theta_0 \neq \theta^\*$, as opposed to a sequence of nulls $\theta_0 = \theta^\* \pm 1/\sqrt{T}$ shrinking towards $\theta^\*$ (such as Pitman alternatives studied commonly in the asymptotic statistics literature). Our specified null, in contrast, is just a fixed value that does not depend on $T$. As such, Lemma 1 only states that for a fixed null $\theta_0 \neq \theta$, it is eventually rejected with probability 1 as $T \rightarrow \infty$.
>
> To be clear, the results of Lemma 1 hold even without simulating with optimism (i.e. adding positive bias to the nuisances). The proof of Lemma 1 relies on a strong law of large numbers (SLLN) argument, paired with a simple fact from Shin et. al. (2021) that enables the SLLN to hold in the adaptive data collection setting. We refer the reviewer to Appendix D.1. for additional details.
>
> To make Lemma 1 clearer, we will emphasize these two points in our manuscript. In particular,
> * We will emphasize that the null is fixed in Lemma 1, and provide additional clarifying remarks beneath Lemma 1 in the text.
> * We will add a remark below Theorem 2 that emphasizes that simulating with optimism is primarily for type I error control, rather than other theoretical guarantees. We will explicitly refer to Lemma 1 in order to make this point clear.
>
> ***Applicability to Other Experiment Designs***:
> Our method does indeed apply to standard randomized designs. If the design has a constant sampling policy across time, our approach is equivalent to randomly sampling observations based on the policy's assignment probability, then generating samples according a gaussian distribution with the estimated mean and variance of the target arm. In this simple case, our approach is a simulated version of the standard Wald confidence interval, and thus has the same asymptotic guarantees.
>
> We emphasize that our approach is intended as a computational approach for settings where *existing approaches for fixed-time inference are not possible*. In studies that do not involve adaptive sampling schemes, such as simple randomized designs mentioned by the reviewer, one has no need to simulate trajectories due to the independence between observations. As such, standard bootstrap methods  for constructing confidence intervals and hypothesis tests apply due to the interchangeability among observations obtained from sampling the same arm. In such a setting, our method should not be used, and one should opt for the plethora of available techniques for this setting such as bootstrapped inference or semi-parametric inference via AIPW [2].
>
> In settings where data is adaptively collected, such as the focus of our work, these standard bootstrap methods fall short. Due to adaptive sampling schemes, an observation at time $t$ influences the observations from times $t+1,...,T$, resulting in dependent observations. As a result, one cannot sample with replacement from samples of the same arm to form bootstrap confidence intervals and hypothesis tests. Our procedure can be seen as mimicking the bootstrap, but instead of sampling with replacement from the observed data, we construct simulated trajectories using the procedure outlined in our manuscript.
>
> **To make this point salient, we will add this discussion to our conclusion to specify which designs our approach is best suited for.**
>
> We refer the reviewer to the discussion on "Difference in Applications with Reweighing/Martingale Methods" in the rebuttal for Reviewer Us9g for a detailed discussion on how our approach differs from existing methods for inference in adaptively collected data. In particular, we note that existing methods cannot provide meaningful inference for Examples 1 and 2 in our work; in contrast, our approach provides valid inference that significantly improves upon the option of anytime-valid inference.
>
> For empirical results regarding sampling schemes beyond Examples 1,2, and 3 of our work, we refer to the rebuttal of Reviewer PfWo. We provide empirical results regarding coverage, interval width, and MSE for Thompson sampling (TS) *without modifications such as clipping and batching*. Our results demonstrate that at least empirically, our approach works well (small confidence intervals, correct coverage) on adaptive sampling schemes beyond the examples for which we prove valid type I error control. In future work, we plan to provide a more general condition for validity across a wider scheme of experiment designs. **We note that these TS experiment results will be added to our appendix as an example of other designs beyond the ones discussed in Section 3**.
>
> **Knowledge of the Sampling Scheme**:
> In contrast to experiments where the sampling scheme (i.e. propensity scores) are fixed in advance, *knowledge of the experiment policy is a standard assumption for after-study inference in adaptive experiments*. In fixed-design settings, propensity scores can be estimated due to being fixed across time (i.e. each arm selection event $\mathbf{1}[A_t = a]$ corresponds to an independent random variable, resulting in $T$ observations for estimating propensity scores). In adaptive settings, where propensities can change across time, we only observe a single observation from each time-varying sampling scheme $\pi_t$: the indicator variable $\mathbf{1}[A_t = a]$. As a result, without further restrictions on the sampling scheme, the exact sampling scheme cannot be estimated, and is assumed to be known across all fixed-time inference approaches on adaptively collected data (to the best of our knowledge).
>
> For a more detailed discussion regarding this point, we refer the reviewer to the "Knowledge of the Experiment Policy" section in the rebuttal for Reviewer Us9g.
>
>
> **Additional Clarifications, Suggestions, and Fixes**
> *  The proposed method is not anytime-valid. As stated in Theorem 1, our approach is a fixed-time test, where $T$ is specified in advance.
> * Algorithm 1 is written correctly, and $\theta_0$ is not necessarily $\hat\mu_1$. Rather, $\theta_0$ is an input to the algorithm that specified the point null to be tested. We will make this clear by adding a remark under Algorithm 1 that specifies the role of the inputs.
> * As the reviewer notes, our approach does not directly apply to settings where one has access to contexts $Z_i$. We plan to investigate this approach in future work and note that Bibaut et. al. (2024) propose a tentative approach (without theoretical guarantees). We will highlight this by adding a comment in our related works section and conclusion.
> * We will add more clarify to the sentence in line 34. In particular, we plan to say:
>
> "standard hypothesis tests and confidence intervals are not guaranteed to provide their nominal error control (e.g., contain the true target of inference with a pre-specified desired error probability) due to the dependence across observations induced by adaptive sampling."
>
> We hope that this clarification, along with the additional remarks discussed in "Applicability to Other Experiment Designs" above, makes this sentence clearer. We thank the reviewer for noting this, and hope this resolves any confusion.
>
> * We apologize for the confusion in deriving Equation (27). For intuition, the numerator is a result of two separate applications of the central limit theorem (with the terms in the indicator following from a strong law of large numbers argument). The denominator serves as a normalizing constant based on the sample size (as determined by the indicator). We will expand on its derivation in the appendix to make this result clear.
>
> We thank the reviewer for raising these points. The suggested edits improve our manuscript to be more conceptually clear, and we will incorporate all clarifications mentioned above. **We hope our responses have addressed your key concerns and would be grateful if you would consider a higher score in light of these clarifications.** Please feel free to ask further questions if any remain!
>
>
> [1] J. Shin, A. Ramdas, and A. Rinaldo. On the bias, risk and consistency of sample means in multi-armed bandits, 2021.
>
> [2]V. Chernozhukov, D. Chetverikov, M. Demirer, E. Duflo, C. Hansen, W. Newey, and J. Robins. Double/debiased machine learning for treatment and structural parameters, 2018.
>
> [3] A. Bibaut and N. Kallus. Demystifying inference after adaptive experiments, 2024.

---

> > ### Comment · Reviewer_4paL · 2025-08-03
> >
> > I thank the authors for their efforts. Most of my concerns have been addressed.

---

> > > ### Author Response · Authors · 2025-08-05
> > >
> > > We are glad that the reviewer found our clarifications and modifications clear, and their concerns addressed. We would be grateful if you would consider a higher score in light of these clarifications! Please feel free to ask any further questions if any remain.

---

### Official Review · Reviewer_DHzj · 2025-06-29

**Clarity:** 3
**Significance:** 3
**Originality:** 3
**Rating:** 5
**Confidence:** 4

**Summary:**

This paper considers the task of statistical inference after adaptive sampling, and proposed a simulation-based approach.  The goal is to overcome two limitations of prior approaches:  First, approaches based on asymptotic normality require restrictions on experimental design (e.g., requiring some lower-bound on the probability of sampling each arm at each step), and approaches based on non-asymptotic anytime-valid inference techniques can lead to weak power in practice. The proposed approach uses a simulation procedure to approximate the distribution of the test statistic under the null.  A core technical innovation is the use of "simulation with optimism" for testing hypotheses related to a single arm, whereby positive noise is added to the estimates of the value of other arms.

**Questions:**

To save the authors time - I have no particular questions at this point, and a detailed response to my comments is not required during rebuttal.  I will reassess if necessary after seeing the reviews of others.

**Ethical Concerns:**

["NO or VERY MINOR ethics concerns only"]

**Final Justification:**

I did not see any reason to change my score.

**Limitations:**

Yes

**Quality:**

3

**Strengths And Weaknesses:**

Overall, I found this paper to be broadly well-written, with a very clear motivation and novel / original technical contribution.  In more detail:

1. (Strength) In terms of clarity and significance, the examples of adaptive sampling designs was extremely helpful for understanding the motivation, particularly as regards the lack of conditional positivity.  See e.g., Examples 1, 2, 3.  If possible, I might have even given these examples earlier on, to help clarify the reason why standard reweighting approaches are not feasible under these designs.
2. (Strength) In terms of originality and technical contribution, I appreciated the core technical contribution of adding positive biases to nuisances in order to maintain Type 1 error control.  Remark 2 is helpful for clarifying why some novel approach is needed, even if nuisances are estimated at parametric rates, and I found the proposed solution to be practical and insightful.
3. (Strength) I appreciated the simulation study and the real-world results, particularly for the intuition they provide regarding where the benefits are most likely to accrue (e.g., in the case where inference is conducted on the worst arm).
4. (Weakness) I think the paper might benefit from some restructuring (up to the authors) to help clarify earlier on what "simulation with optimism" means, and the specific setting it is being applied to.  In particular, I was initially confused by the fact that the setup is very general up until page 5, and it was only in Section 3.2.1 that I started to understand that this approach applies in fairly specific settings: Theorem 1 in particular is a result focused specifically on these three example designs, and specifically for testing a point null for the mean of a single arm.  I think this specific setting could use more discussion upfront to clarify where the results apply.

I would also like to pre-emptively defend (in case other reviewers comment on it) the decision to focus on asymptotic control (see Remark 1). While non-asymptotic inference is conceptually appealing, I am familiar with it's limitations regarding power and flexibility in practice.

---

> ### Author Rebuttal · Authors · 2025-07-29
>
> We thank the reviewer for their encouraging review of our work. In particular, we are glad that the reviewer found our manuscript well-written and clear, our method both novel and technically sound, and our empirical results compelling.
>
> We hope that the modifications to our work (particularly the additional discussion/remarks on general sampling schemes discussed on rebuttals for Reviewers 4paL and PfWo) help clarify the weaknesses mentioned by the author. In particular, we provide additional details in new remarks that (i) emphasize which designs to which our work applies and (ii) where standard methods should be used over our approach. In addition, we will emphasize in Section 2 (setup, problem statement) that our approach is intended for designs violating conditional positivity, and refer to Examples 1 and 2 earlier.

---

### Official Review · Reviewer_PfWo · 2025-07-01

**Clarity:** 4
**Significance:** 3
**Originality:** 3
**Rating:** 5
**Confidence:** 4

**Summary:**

This paper considers the problem of doing frequentist hypothesis testing and constructing frequentist confidence intervals after using bandit-driven adaptive experimental design. The basic idea is to run simulations of the adaptive experimental procedure under the null hypothesis and build an empirical distribution of the test statistic, using this to construct a CI (which can also be used as a hypothesis test). The wrinkle is that simulating under the null distribution requires knowledge of the arm values of all other arms, since these influence the adaptive experimental procedure. This paper first shows that using a raw mean as an estimate in the simulator is not suitable, because it does not lead to the right limiting distribution. The paper then proposes a correction to the mean of arm $i$ to be used in the simulator given by $\\log\\log N\_T(i)/\\sqrt{N\_T(i)}$. This correction ultimately derives from the law of the iterated logarithm. They prove that this correction, combined with simulation inference as before, leads to the correct control of Type I error (for certain experimental design schemes).

**Questions:**

- the real world data was collected with batch Thompson Sampling. Yet TS is not covered by Examples 1, 2, 3 as far as I can see. Have you explored providing a proof of Theorem 1 for TS?

**Ethical Concerns:**

["NO or VERY MINOR ethics concerns only"]

**Final Justification:**

Updated my rating in light of clarification in the rebuttal phase, particularly about how this would relate to both the TS policy used to collect the real data and empirical evidence under a "raw" TS policy

**Limitations:**

Yes

**Paper Formatting Concerns:**

No concerns

**Quality:**

3

**Strengths And Weaknesses:**

# Strengths
- writing quality is excellent with clear explanation throughout, notation is defined and consistent
- examples are provided to ground the reasoning
- resulting procedure is simple to apply with limited assumptions on the underlying distribution
- experiments show that confidence interval widths tend to be smaller when using this approach, with some significant reductions for arms that were chosen less often

# Weaknesses
- Theorem 1, the main result, explicitly refers to Examples 1, 2, 3. Thus, this result is not entirely general to any experimental design set-up and would need to be re-proven. This is really the main drawback as far as I can make out.
- Further empirical investigation could have been conducted to understand whether Theorem 1 really seems like a general result. For instance, repeating the simulated data experiment with a wide range of popular experimental design strategies.
- The resulting algorithm is much more costly than some conservative yet simple approaches. Yet in the setting we consider, in which data collection may have been very costly, I don't over-weight this issue.

---

> ### Author Rebuttal · Authors · 2025-07-29
>
> We thank the reviewer for their detailed comments and questions. In particular, we are glad that the reviewer found our writing clear, examples grounded, and empirical results compelling. We hope to address any questions and concerns below and have added new experiment results that address the reviewer's concerns.
>
> ***Applicability to General Experiment Designs***
> As the reviewer notes, our work provides the theoretical guarantees of Theorem 1 under Examples 1,2,3, and does not directly apply to the arbitrary general designs. Below, we discuss the generality of our examples (particularly Example 3) and provide additional empirical results for designs not described by our examples.
>
> **Additional Details on our Examples.** While Examples 1 and 2 describe explicit sampling schemes (explore-then-commit and one particular version of UCB respectively), Example 3 consists of a broad range of experimental designs. In particular, Example 3 corresponds to $\textbf{all}$ regret-optimal (i.e. reward-maximizing) algorithms modified with a positive lower bound on its selection probabilities, provided that the experiment has a unique best arm. To connect regret optimality with the conditions of Example 3, note that the minimum possible regret scales on the order of $\log(T)$ as $T \rightarrow \infty$ (Lattimore and Szepesvari, 2020). As such, a bandit algorithm is only regret optimal if and only if there exists a fixed constant $c < \infty$ such that suboptimal arms are pulled at the rate $c/t$ at each timepoint due to $\sum_{t=1}^T 1/t \approx \log(T)$. Example 3 therefore captures any reward-maximizing scheme known to be theoretically optimal (e.g. UCB variants such as MOSS-UCB [2], Thompson Sampling [3]) modified with clipping for arm instances with a unique best arm. We note that these clipped schemes are often used for empirical studies in the literature, with applications from political science surveys [4] to digital health interventions [5], where one typically assumes that a best arm exists. Thus, while our examples may seem limited, many real-world applications use the experiment designs covered in Examples 1, 2, and 3.
>
> As noted by the reviewer, Theorem 1 does not provide explicit type I error guarantees for examples that lie outside the class of designs specified by our examples. This is due to the poor behavior of even the simplest of bandit algorithms in settings where the arm gaps are small (relative to the sample size). For example, to characterize the behavior of standard Thompson sampling (i.e. no batching and clipping), existing works ([6,7]) show that the limits of arm pulls even in the two-armed case correspond to systems of differential equations, making inference guarantees difficult to obtain. We plan on pursuing guarantees for a more general class of designs in future work, and believe that this would be out-of-scope for this submission, which introduces our method and proves error guarantees for designs (Examples 1 and 2) where previous methods do not apply.
>
> **To make our contributions and challenges clear, we plan to summarize the discussion above as a remark under Theorem 1. We thank the reviewer for their questions and comments.**
>
> **Additional Experiments Beyond Our Examples**
> Following the reviewer's suggestions for empirical evaluation, we test Beta-Bernoulli Thompson sampling *without batching and clipping*, a design that *does not satify the requirements of any of our examples*. We note that for all baselines other than the anytime-valid method, type I error is not guaranteed. Like Figure 2, we track coverage, CI width, and MSE. We provide the results for inference on the best arm and the worst arm for a three-armed experiment with $\mu = [0.45, 0.5, 0.55]$. We set $\alpha = 0.1$ and $T=400$. We note that we set the horizon to a relatively small number to avoid issues with zero propensity scores (which would make our baselines produce trivial results). We report our results with 100 simulations of our method in the tables below, where 1 standard deviation is included in the $\pm$ term.
>
> |                   | Coverage        | CI Width          | MSE            |
> |-------------------|-----------------|-------------------|----------------|
> | Anytime-Valid     | 0.98 ± 0.01     | 0.23 ± 0.002      | 0.006 ± 0.000  |
> | Stable Weights    | 0.89 ± 0.03     | 0.14 ± 0.002      | 0.004 ± 0.000  |
> | Two Point Weights | 0.84 ± 0.04     | 0.16 ± 0.002      | 0.006 ± 0.000  |
> | Simulation        | 0.87 ± 0.03     | 0.13 ± 0.002      | 0.004 ± 0.000  |
>
> Table 1: Inference on best arm mean $\mu_3 = 0.55$.
>
>
> |                   | Coverage        | CI Width          | MSE            |
> |-------------------|-----------------|-------------------|----------------|
> | Anytime-Valid     | 0.88 ± 0.03     | 0.31 ± 0.001      | 0.005 ± 0.000  |
> | Stable Weights    | 0.69 ± 0.05     | 0.18 ± 0.001      | 0.005 ± 0.000  |
> | Two Point Weights | 0.69 ± 0.05     | 0.21 ± 0.001      | 0.006 ± 0.000  |
> | Simulation        | 0.88 ± 0.03     | 0.21 ± 0.001      | 0.004 ± 0.000  |
>
> Table 2: Inference on the worst arm mean $\mu_1 = 0.45$.
>
>
> Our results demonstrate that relative to the baselines, the simulation procedure provides *tighter confidence intervals without sacrificing type I error*. When conducting inference on the best arm, our approach achieves close to the nominal coverage level, while producing the tightest confidence interval among all methods. For the worst arm, we see that stable weights and two point weights, inference approaches based on martingale central limit theorems (MCLT), provide relatively small confidence intervals, but suffer from significant undercoverage. In contrast, both our anytime-valid baseline and our simulation approach result in coverage close to nominal levels. Between these two methods, our simulation approach produces much smaller confidence intervals, demonstrating its benefits empirically.
>
> Thus, while we may currently lack theoretical guarantees for general sampling algorithms, our simulation approach retains its competitive inference performance without sacrificing type I error empirically.
>
> **We plan to add these results in our appendix, and will discuss more general conditions for valid type I error as a direction for future work in our conclusion.**
>
> ***Validity Under Batch Thompson Sampling Experiment***
> We would like to clarify that Example 3 *indeed* covers the batched, clipped TS experimental data used for our real-world case study. To show this, note that:
> * Unique best arm: The authors who collected the data from the study [4] explicitly state that they assume that there exists a unique best arm achieving the highest mean. In Appendix B.1., the authors state "For estimation and hypothesis testing, we have assumed that there is a unique best arm." Empirical results in Figures 3 and 4 of [4] (right-side figures correspond to our data) suggest that this is true.
> * Clipped design: The authors of [4] discuss the sampling scheme on pg. 19 of their work:
>     "we implemented a composite adaptive-static design: for each type of ballot measure, subjects were assigned treatment according to Thompson sampling with 90\% probability, and according to balanced simple random assignment with 10\% probability."
> As such, this satisfies the clipping requirement, where $\gamma = 0.1$ as defined in Example 3.
> * Batched TS satisfies regret-optimality: Existing work [8] has shown that batched TS preserves the problem-specific regret optimality $O(\log T)$. Following the argument outlined above (regarding our sampling condition and regret optimality), it follows that there exists a $c < \infty$ such that $\mathbb{P}(A_t \neq a^*) \leq c/t$.
>
> **Therefore, the dataset used for our real-world case study is collected under a setting that satisfies the conditions of Example 3. To make this clear, we will add this justification to the Appendix (with a reference in the main text) to justify the use of our algorithm on this dataset.**
>
> We thank the reviewer for raising these points. We will incorporate these clarifications (as mentioned above) into our final draft, and hope that this addresses any questions or comments the reviewer may have. **We hope our responses have addressed your key concerns and would be grateful if you would consider a higher score in light of these clarifications and additions.** Please feel free to ask further questions if any remain!
>
> [1] Bandit Algorithms. T. Lattimore, and C. Szepesvari. (2017)
>
> [2] Anytime optimal algorithms in stochastic multi-armed bandits. Rémy Degenne and Vianney Perchet. (2016)
>
> [3] Further Optimal Regret Bounds for Thompson Sampling. Shipra Agrawal and Navin Goyal. (2013)
>
> [4]Adaptive experimental design: Prospects and applications in political science. Molly Offer‑Westort, Alexander Coppock, Donald P. Green. (2021)
>
> [5] Did we personalize? Assessing personalization by an online reinforcement learning algorithm using resampling. Susobhan Ghosh, Raphael Kim, Prasidh Chhabria, Raaz Dwivedi, Predrag Klasnja, Peng Liao, Kelly Zhang, Susan Murphy. (2023)
>
> [6] Diffusion Approximations for Thompson Sampling. Lin Fan and Peter Glynn. (2021)
>
> [7] Diffusion Asymptotics for Sequential Experiments. Stefan Wager and Kuang Xu. (2021)
>
> [8] Batched Thompson Sampling. Cem Kalkanlı and Ayfer Özgür. (2021)

---

> > ### Comment · Reviewer_PfWo · 2025-08-04
> > **Thank you for your response**
> >
> > I would like to sincerely thank the authors for a detailed response to my comments and questions. I particularly appreciate your clarification about the scope of Example 3 in the manuscript. It was not clear that the TS policy used when collecting the real data actually falls within the scope of the proof in the paper. This indeed strengthens the paper. I also appreciate the addition of a new experiment that falls *outside* the scope of the proofs, yet provides empirical evidence that the simulation-based approach proposed in the submission still operates effectively. Overall- clarification of these points in the updated version of the paper would be great.
> >
> > I will update my score on the paper accordingly.

---

> > > ### Author Response · Authors · 2025-08-05
> > >
> > > We are glad that the reviewer found our clarifications beneficial to our work - we will include all mentioned changes in the final manuscript. Thank you for your revised score!

---

### Note · Authors · 2025-08-14

We thank all reviewers for their thoughtful feedback—your input has already improved and strengthened out work.

Reviewers highlight that our work fills an important methodological and practical gap: our simulation-based method provides valid inference across common experiment designs that violate assumptions necessary for standard inference procedures (PfWo, DHzj, 4paL, Us9g), avoids stringent assumptions on the underlying distribution (PfWo, 4paL), and achieves strong empirical performance relative to existing methods in the literature (PfWo, Us9g). Furthermore, reviewers generally found our work well-written and clear (PfWo, DHzj), with examples throughout that ground both the theoretical and practical significance of our work.

Building on these strengths, to the best of our knowledge, our approach of simulating with optimism is the *first approach applicable to the designs studied in our work* (to the best of our knowledge) for *fixed-time, after-study* inference. While methods such as anytime-valid inference offer valid asymptotic error control in the settings we consider, our approach both offers both error control and *strong* practical performance, mitigating power issues that often limit existing approaches.

We thank the reviewers once again for their valuable input, and all discussed changes will be incorporated into the final manuscript.

---

### Decision · Program_Chairs · 2025-09-17

**Decision:**

Accept (poster)

**Comment:**

This paper develops methods for confidence intervals after using bandit-driven adaptive experimental design.

This paper has relatively a good writing quality and clarity along with good examples. The resulting procedure is reasonably simple, and the experimental results are promising. There was one suggestion from a review regarding restructuring the paper so that the meaning of 'simulation with optimism' is clear up front. Please take it into account for the final version.